

# 1  Comprehensive evaluation of typical planetary boundary
# 2  layer (PBL) parameterization schemes in China. Part I:
# 3  Understanding expressiveness of schemes for different
# 4  regions from the mechanism perspective

Wenxing Jia[1], Xiaoye Zhang[1,2*], Hong Wang[1], Yaqiang Wang[1], Deying Wang[1], Junting Zhong[1],
Wenjie Zhang[1], Lei Zhang[1], Lifeng Guo[1], Yadong Lei[1], Jizhi Wang[1], Yuanqin Yang[1], Yi Lin[3]
[1]State Key Laboratory of Severe Weather & Key Laboratory of Atmospheric Chemistry of CMA,
Chinese Academy of Meteorological Sciences, Beijing, 100081, China
[2]Center for Excellence in Regional Atmospheric Environment, IUE, Chinese Academy of Sciences,
Xiamen, 361021, China
[3]Key Laboratory for Mesoscale Severe Weather, Ministry of Education, and School of Atmospheric
Sciences, Nanjing University, Nanjing, China
Correspondence to: X. Zhang (xiaoye@cma.gov.cn)





**Abstract.** The optimal choice of the planetary boundary layer (PBL) parameterization scheme is of
particular interest and urgency to a wide range of scholars, especially for many works involving
models. At present, there have been many works to evaluate the PBL schemes. However, little
research has been conducted into a more comprehensive and systematic assessment of the
performance capability of schemes in key regions of China, especially when it comes to the
differences in the mechanisms of the schemes themselves, primarily because there's scarcely
sufficient observational data, computer resources, and storage support to complete the work. In
addition, there are many factors that influence the selection of schemes, such as underlying surface,
initial and boundary conditions, near-surface layer scheme, horizontal/vertical resolution, etc. In
this Part (i.e., Part I), four typical schemes (i.e., YSU, ACM2, BL and MYJ) are selected to
systematically analyze and evaluate near-surface meteorological parameters, PBL vertical structure,
PBL height (PBLH), and turbulent diffusion in five key regions (i.e., North China Plain, NCP;
Yangtze River Delta, YRD; Sichuan Basin, SB; Pearl River Delta, PRD and Northwest Semi-arid,
NS) of China in different seasons (i.e., January, April, July and October). The differences in the
simulated 2-m temperatures between the nonlocal closure schemes are mainly affected by the
downward shortwave radiation, but to compare the nonlocal closure schemes with the local closure
schemes, the effect of sensible heat flux needs to be further considered. In terms of temporal
variation, the simulated results for July are better than the other three months, and the simulated
results for nighttime are better than daytime. In terms of regional distribution, the temperature at
stations with higher elevation is easily underestimated, while overestimated with lower elevation.
The variation of relative humidity corresponds to temperature. The 10-m wind speed is under the
influence of factors like the momentum transfer coefficient and the integrated similarity functions
at night. The wind speeds are more significantly overestimated in the plains and basin, while less
overestimated or even underestimated in the mountains, as a result of the effect on topographic
smoothing in the model. Moreover, the overestimation of small wind speeds at night is attributable
to the inapplicability of the Monin-Obukhov similarity theory (MOST) at night. The model captures
the vertical structure of temperature well, while the wind speed is outstandingly overestimated
below 1000 m, largely because of the turbulent diffusion coefficient (TDC). The difference between
the MOST and the mixing length theory, PBLH and Prandtl number is cited as the reason for the
difference between the TDC of the YSU and ACM2 schemes. The TDCs of the BL and MYJ
schemes are affected by the mixing length scale, which of BL is calculated on the basis of the effect
of buoyancy, while MYJ calculates it with the consideration of the effect of the total turbulent
kinetic energy. The PBLH of the BL scheme is better than the other schemes because of the better
simulation results of temperature. The difference in the PBLH by the YSU and ACM2 scheme
mainly comes from the Richardson number and the jagged PBLH of the MYJ scheme is due to the
coarse vertical resolution and the threshold value.



In general, to select the optimal scheme, it is necessary to offer different options for different regions
with different focuses (heat or momentum). (1) Temperature field. The BL scheme is recommended
for January in the NCP region, especially for Beijing, and the MYJ scheme is better for the other
three months. The ACM2 scheme would be a good match for the YRD region, where the simulation
differences between the four schemes are small. The topography of the SB region is more complex,
but for most of the areas in the basin, the MYJ scheme is proposed, but if more stations outside the
basin are involved, the BL scheme is recommended. The MYJ scheme is applied to the PRD region
in January and April, and the BL scheme in July and October. The MYJ scheme is counselled for
the NS region. (2) Wind field. The YSU scheme is recommended if the main concern is the near-
surface layer, and the BL scheme is suggested if focusing on the variation in the vertical direction.
The Part II will analyse and evaluate the factors that may influence the choice of the schemes and
the results of model. The final evaluation of the parameterization scheme and uncertainties will lay
the foundation for the improvement of the modules and forecasting of the GRAPES_CUACE
regional model developed independently in China.
**1 Introduction**
The planetary boundary layer (PBL) is the part of the troposphere that is directly influenced by the
force of the earth's surface with an hour or less timescale(R. B. Stull, 1988). Parameterization is the
determining factor in the predictive accuracy and skill as it determines key aspects of simulated
weather(Bauer et al., 2015; Williams, 2005). In numerical weather prediction, meagre
computational resources limit the resolution of the model. Following this reason, physical processes
cannot be resolved by the model in that the spatial scales are smaller than the model grid distance.
The physical module in the model that characterizes small scales relative to the model resolution is
called the sub-grid physical process parameterization scheme(Zhou et al., 2017). As a typical sub-
grid parameterization scheme, the spatial scale of turbulence is limited by the PBL height (PBLH)
and cannot be resolved by mesoscale weather prediction models and macroscale global climate
models with horizontal grid distances of magnitude of ~10 km and ~100 km. Therefore, the physical
module in the model that describes the effect of sub-grid turbulence on resolvable atmospheric
motion is called the PBL parameterization scheme. Even in the high resolution large-eddy
simulation (LES), small-scale turbulence requires parametric closure to characterize the role of sub-
grid turbulence(Deardorff, 1980). The PBL parameterization scheme controls the evolution of
momentum, heat, water vapor, and mass within the PBL, and the evolution of these parameters is
particularly affected by the turbulent diffusion coefficients (TDCs)(W. Jia and Zhang, 2021;
Nielsen-Gammon et al., 2010; Oke et al., 2017). Depending on the turbulence closure method, the
PBL parameterization schemes can be divided into three main categories: nonlocal closure schemes,





local closure schemes, and hybrid nonlocal-local closure schemes, and the above schemes have their
own advantages and disadvantages(Cohen et al., 2015; Hu et al., 2010; Wenxing Jia and Zhang,
2020; Xie et al., 2012).
Since the early 1980s, the vertical diffusion scheme based on local gradients of wind and potential
temperature (i.e., local K-theory) has been applied in the National Centers for Environmental
Prediction (NCEP). However, as pointed out by many scholars, this scheme has many deficiencies,
of which the most critical is that the mass and momentum transport within the PBL is mainly
accomplished by the large-scale eddies besides the local small-scale eddies(Roland B. Stull, 1984;
Wyngaard and Brost, 1984). Therefore, the new scheme developed later incorporates a counter-
gradient flux term to characterize the turbulent transport processes in large-scale eddies, such as
Medium-Range Forecast (MRF) scheme (i.e., nonlocal closure) (Hong and Pan, 1996; Troen and
Mahrt, 1986). This scheme has also been commonly used in China's self-developed
Global/Regional Assimilation and PrEdiction System (GRAPES) model because of its
computational simplicity and its ability to produce plausible results under typical atmospheric
conditions (Ma et al., 2021). Nevertheless, the MRF scheme has gradually shown some
shortcomings, the most typical being that when the wind speed is strong, the resulting mixing is too
strong and thus the PBLH is too high to be realistic(Mass et al., 2002; Persson et al., 2001). To
overcome this critical problem, one of the most commonly used and popular PBL parameterization
scheme has been introduced, which is the Yonsei University (YSU) scheme(Hong et al., 2006). YSU
scheme adds an additional entrainment term to the MRF scheme for explicitly calculating the
entrainment process of heat and momentum fluxes(Noh et al., 2003). It is still unclear why this
scheme is popular among scholars, either because it gives the best simulation results or simply
because the code of this scheme is 1, which is more convenient for the model setting. To be contrast,
a newer scheme, as a nonlocal scheme of the same series, has been developed that further considers
the issue of gray-zone of sub-grid scale turbulence, but this scheme has been rarely used and
evaluated (Hong and Shin, 2013).
Repairing the defects of local K-theory is possible by developing nonlocal closure schemes on the
one hand, and higher-order local closure method on the other hand. The most representative is the
higher-order closure scheme of the M-Y series proposed by Mellor and Yamada, such as Mellor-
Yamada-Janjic (MYJ) scheme and Mellor-Yamada Nakanishi and Niino Level 2.5/3
(MYNN2/MYNN3) scheme(Janjić, 1990, 1994; Mellor and Yamada, 1974, 1982; Nakanishi and
Niino, 2004). The higher-order closure schemes are capable of representing a well mixing PBL
structure, however, these schemes are computationally more expensive due to the addition of a
prognostic turbulent kinetic energy (TKE). In addition to the widely used local closure schemes of
the M-Y series, there is another local closure scheme that has been evaluated extensively. This
scheme is the Bougeault and Lacarrere (BL) scheme(Bougeault and Lacarrere, 1989), but there are


several differences between the BL and M-Y schemes. (1) In the parameterization of the turbulent
heat flux, an additional counter-gradient flux term is taken into account in the convective PBL, but
this counter-gradient term is different from that in the nonlocal closure scheme, which is a constant
($= 0.7 \cdot 10^{-5}\ K\ cm^{-1}$) in the BL scheme. (2) The turbulent diffusion coefficient in the BL scheme
is calculated similarly to the M-Y schemes, but the stability functions and mixing length are different
from M-Y schemes.
In addition to the typical nonlocal closure schemes and local closure schemes, there are also hybrid
nonlocal-local closure schemes, typically represented by the Asymmetric Convective Model version
2 (ACM2) scheme. ACM2 scheme operates based on the development of ACM1 that is modified
based upon the Blackadar convective model(Blackadar, 1962). The upward transport within the PBL
is mainly by buoyancy, which is transmitted upward from the lowest level to other levels, while
downward is transported level-by-level(Pleim, 2007). The deficiency of the ACM1 scheme is that
upward transport is not better represented when the vertical resolution of the model increases. In
response to compensating for the shortcomings of the ACM1 scheme, the ACM2 scheme adds level-
by-level transport to the upward level. The ACM2 scheme have the highest universality and was
most suitable for the study of meteorological elements in desert region(Meng Lu et al., 2018; Wang
et al., 2017).
At present, a total of 12 PBL parameterization schemes have been developed and evaluated in the
currently popular mesoscale Weather Research and Forecasting (WRF) model. The continuous
improvement of numerical simulation techniques brings opportunities for the update and
development of PBL parameterization schemes. Many scholars hope that by comparing the PBL
parameterization schemes, they can select one scheme that better reflects the changes in
meteorological parameters (e.g., temperature, relative humidity and wind speed/direction),
pollutants and the structures of the PBL. Recent review studies have shown that although many
studies on the evaluation and comparison of PBL parameterization schemes have been undertaken,
there is still no uniform conclusion on which PBL parameterization scheme performs best(Wenxing
Jia and Zhang, 2020). Moreover, most of the evaluation work on PBL parameterization schemes is
done for individual cases or a particular region(Avolio et al., 2017; Diaz et al., 2021; Falasca et al.,
2021; Ferrero et al., 2018; He et al., 2022; Shen et al., 2022). In spite of those, simulation results for
the PBL parameterization schemes are more uniform: (1) the simulation of temperature is better
than that of relative humidity, and the simulation of wind speed and direction is worse. (2) The
simulation results of the nonlocal closure scheme are better under unstable conditions, while the
local closure scheme for stable conditions. However, these general conclusions are open to
speculation and debate(Wenxing Jia and Zhang, 2020). Many previous studies have been biased
towards the assessment of basic meteorological parameters, of course, which is the basic work. Due
to the indirect output of TDC by the model, there are fewer relevant studies to investigate the impacts



of turbulent diffusion on meteorological parameters. Moreover, turbulent diffusion is the key factor
to control the vertical mixing of momentum and scalars within the PBL. Even if there is not enough
turbulence observation data, it can be further analyzed and discussed according to the simulation
results. Aimed at remedying the current research deficiencies, this study first selects four typical
boundary layer parameterization schemes (nonlocal scheme: YSU, local scheme: MYJ and BL,
hybrid nonlocal-local scheme: ACM2) for five typical regions (NCP, YRD, SB, PRD and NS) in
China, and then assesses the performance capability of different PBL parameterization schemes in
different regions. The reasons for the differences in performance of meteorological parameters
between observation and simulation are illustrated in terms of temporal and regional variability.
Then, the mechanistic implications behind the differences are explored between schemes. In
addition, we further carry out the comparative analysis of the vertical structure of the PBL, turbulent
diffusion and the PBLH. The first part of this study (i.e., Part I) aims to be able to have a qualitative
and quantitative assessment of the PBL parameterization schemes in different regions for other
researchers to use as a reference when doing simulation studies. The second part (i.e., Part II)
focuses on the analysis of some uncertain factors that may affect the model simulation results,
chiefly including: the influence of meteorological initial and boundary conditions, underlying
surface (mainly considering the impact of urban and water bodies), near-surface layer (N-SL)
scheme (the PBL and N-SL schemes must match each other), the effect of model version update,
the influence of regional horizontal and vertical resolution, etc. We hope that we can dissect the
effect of uncertainties from some aspects that we are concerned about.
**2 Data and methods**
**2.1 Data**
***Hourly meteorological observation data.*** The China Meteorological Administration (CMA) has
over 2400 automatic weather stations (AWSs), and the stations record variables such as temperature,
relative humidity, pressure, wind speed, wind direction and precipitation amount. In the NCP, YRD,
SB, PRD, NS regions, 576 stations, 455 stations, 341 stations, 128 stations, and 55 stations have
been selected, respectively (illustrated by gray cross in Fig. 1b-f). Observational data for four
months January, April, July and October 2016 have been selected and comparatively analyzed.
***L-band radiosonde observation data.*** A total of 120 observation stations are equipped with L-band
radiosonde systems in China, which provide fine-resolution (1 Hz, and the rise rate is ~6 m s$^{-1}$)
vertical profiles of temperature, relative humidity and wind speed and direction three times (08:00,
14:00 and 20:00 Beijing Time, BJT) a day (illustrated by red triangle in Fig. 1b-f). Four sounding
stations have been selected for each region, including different underlying surface conditions as
much as possible. In the NCP region, two plain stations (Beijing and Xingtai) and two mountain


stations (Zhangjiakou and Zhangqiu) have been picked. In the YRD region, one station closer to the
ocean (Shanghai), two stations with complex underlying surface (Anqing and Quzhou), and one
plain station (Nanjing) have been opted. In the SB region, three in-basin stations (Wenjiang,
Shapingba and Daxian) and one out-of-basin station (Hongyuan, with an altitude of 3491 m) have
been selected. In the PRD region, two plain stations (Qingyuan and Heyuan) and two stations
(Yangjiang and Shantou) closer to the ocean have been singled out. In the NS region, along the
Qilian mountains, four stations have been chosen, Mazongshan, with an altitude of 1770 m, Jiuquan,
with an altitude of 1477 m, Zhangye, with an altitude of 1460 m, and Minqin, with an altitude of
1367 m.
**2.2 Model settings**
In this study, we adopt the model WRF-ARW (Advanced Research Weather Research and
Forecasting) version 3.9.1 to evaluate the performance of PBL schemes. Long-term three-
dimensional simulation experiments are conducted in 1 month of each season of 2016 (i.e., January,
April, July and October). Seven nested domains (D1, D2, D3, D4, D5, D6 and D7) are defined (Fig.,
1a), with horizontal grid spacings of 75 km (74 × 74 grid cells, 9°N -59°N, 61°E -146°E), 15 km
(281 × 281 grid cells, 13°N -51°N, 77°E -136°E), 3 km (331 × 331 grid cells, 34°N -44°N, 111°E
-124°E), 3 km (316 × 356 grid cells, 26°N -34°N, 114°E -126°E), 3 km (331 × 331 grid cells, 25°
N -33.5°N, 100°E -110°E), 3 km (236 × 301 grid cells, 18°N -25°N, 110°E -119°E), and 3 km (226
× 351 grid cells, 36°N -42.7°N, 94°E -107°E), respectively. Along the vertical direction, 48 vertical
layers are configured blow the top, and the model top is set to the 50 hPa. To resolve the PBL
structure finely, 21 vertical layers are set below 2 km (i.e., the specific setting of vertical levels is σ
= 1.000, 0.997, 0.994, 0.991, 0.988, 0.985, 0.980, 0.975, 0.970, 0.960, 0.950, 0.940, 0.930, 0.920,
0.910, 0.895, 0.880, 0.865, 0.850, 0.825, 0.800). The initial and boundary conditions of
meteorological fields are set up by using the NCEP Global Forecast System (GFS) Final (FNL)
gridded analysis datasets, with a resolution of 1° × 1° (https://rda.ucar.edu/datasets/ds083.2/, last
access: 4 August, 2022). The Moderate Resolution Imaging Spectroradiometer (MODIS) dataset
includes 20 land-use categories(Broxton et al., 2014). The physical parameterization used in the
present model is listed in Table 1.



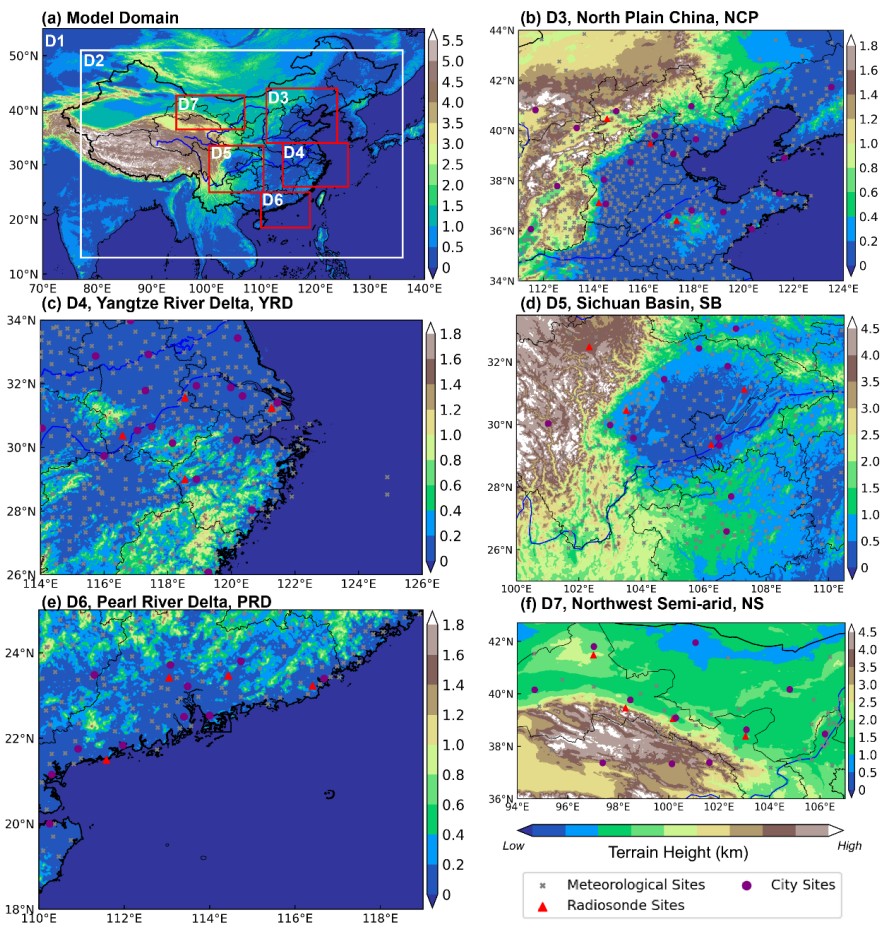

**Figure 1. (a) Map of terrain height in the seven nested model domains. (b-f) Domain 3-7 correspond to the North Plain China (NCP), the Yangtze River Delta (YRD), the Sichuan Basin (SB), the Pearl River Delta (PRD) and the Northwest Semi-arid (NS), respectively. The locations of surface meteorological stations and sounding stations are marked by the gray crosses, red triangles, respectively. The purple dots indicate the major city sites that are our main focus in each region.**

**Table 1. A brief description of the parameterization scheme in the model.**

| Namelist option | Description | Input option | Reference |
|---|---|---|---|
| mp_physics | Morrison double-moment scheme | 10 | (Morrison et al., 2009) |
| ra_lw_physics | RRTMG scheme | 4 | (Iacono et al., 2008) |
| ra_sw_physics | RRTMG scheme | 4 | (Iacono et al., 2008) |
| cu_physics | Grell-3D scheme | 5 | (Grell and Dévényi, 2002) |
| sf_sfclay_physics | MM5 similarity scheme | 1 | (Jiménez and Dudhia, 2012) |
| sf_surface_physics | Noah land surface scheme | 2 | (Chen and Dudhia, 2001) |
| sf_urban_physics | Single-layer UCM | 1 | (Kusaka et al., 2001) |





| | scheme | | |
|---|---|---|---|
| sf_lake_physics | CLM4.5 lake scheme | 1 | (Gu et al., 2015) |
| | YSU scheme | 1 | (Hong et al., 2006) |
| | MYJ scheme | 2 | (Mellor and Yamada, 1982) |
| bl_pbl_physics | ACM2 scheme | 7 | (Pleim, 2007) |
| | BL scheme | 8 | (Bougeault and Lacarrere, 1989) |

All simulations embodied a total of 16 months. The 40 h simulation is conducted beginning from
00:00 UTC of 1d ago for each day (i.e., 492 simulation experiments), the first 16 h of each
simulation is considered as the spin-up period, and results obtained from the following 24 h
simulations are analyzed for the present study.
**2.3 Description of PBL parameterization schemes**
**2.3.1 YSU scheme**
The YSU is a first-order nonlocal scheme with an explicit treatment entrainment process at the top
of the PBL:
$\frac{\partial c}{\partial t} = \frac{\partial}{\partial z}\left[K_c\left(\frac{\partial c}{\partial z} - \gamma_c\right) - \overline{(w'c')}_h\left(\frac{z}{h}\right)^3\right]$ (1)
where $c$ denotes $u$, $v$, $\theta$, and the $\gamma_c = b\frac{\overline{(w'c')}_0}{w_{s0}h}$ is the counter-gradient flux term, which increases
the nonlocal effect due to the large scale turbulence. $z$ and $h$ are the height of a level of the model
and PBLH, respectively. The PBLH is defined by the bulk Richardson number method:
$h = Rib_{cr}\frac{\theta_{va}|U(h)|^2}{g(\theta_v(h) - \theta_s)}$ (2)
where $g$ is the gravity, and $Rib$ is the critical bulk Richardson number, with a value of 0.25 under
stable conditions and 0 under unstable conditions. $\theta_{va}$ is the virtual potential temperature at the
lowest model level, $\theta_v(h)$ is the virtual potential temperature at $h$, $\theta_s$ is the appropriate
temperature near the surface ($\theta_s = \theta_{va} + \theta_T$, $\theta_T$ is the virtual temperature increment). Compared
to the predecessor of the YSU scheme, the entrainment process is additionally treated explicitly (i.e.,
the last term on the right side of the Eq. (1))
Another key variable is $K_c$, which is the turbulent diffusion coefficient (TDC), and can be
expressed based on the Monin-Obukhov similarity theory (MOST) as:
$K_c = \frac{\kappa u_* z}{\phi_c}\left(1 - \frac{z}{h}\right)^2$ (3)
where $u_*$ is the surface frictional velocity and $\phi_c$ is dimensionless function, the expressions for
different stability conditions are:
i. Unstable and neutral conditions:





$\phi_m = \left(1 - 16\frac{0.1h}{L}\right)^{-1/4}$   (4a)
$\phi_h = \left(1 - 16\frac{0.1h}{L}\right)^{-1/2}$   (4b)
ii.      Stable condition:
$\phi_m = \phi_h = \left(1 + 5\frac{0.1h}{L}\right)$   (4c)
The TDC of momentum (i.e., $K_m$) is first calculated in the model, and then the TDC of heat (i.e.,
$K_h$) is calculated, using the Prandtl number (i.e., $Pr = \frac{K_m}{K_h}$). The TDC controls the vertical mixing
process of momentum and scalars within the PBL, and it is crucial that it needs to be accurately
described.
**2.3.2 MYJ scheme**
The MYJ scheme is a one-and-a-half order local closure scheme with a prognostic equation for
turbulent kinetic energy (TKE, $TKE = e = \frac{1}{2}(u'^2 + v'^2 + w'^2)$):
$\frac{\partial \bar{e}}{\partial t} = -\frac{1}{\bar{\rho}}\frac{\partial}{\partial z}\overline{w'p'} - \overline{w'u'}\frac{\partial \bar{u}}{\partial z} - \overline{w'v'}\frac{\partial \bar{v}}{\partial z} - \frac{\partial}{\partial z}\overline{w'e'} + \frac{g}{\theta_v}\overline{w'\theta_v'} - \varepsilon$   (5)
The first term on the right side of Eq. (5) is a pressure correlation term which describes TKE is
redistributed by pressure perturbations, the second and third terms is a shear production/loss term,
the fourth term represents the turbulent transport of TKE, the fifth term describes the buoyant
production/consumption term, and the sixth term represents viscous dissipation of TKE. To close
the TKE equation, the turbulent fluxes must be parameterized. Based on the gradient transport
theory (i.e., K-theory), the turbulent fluxes can be indicated as:
$\overline{w'u'} = -K_m\frac{\partial \bar{u}}{\partial z}$   (6a)
$\overline{w'v'} = -K_m\frac{\partial \bar{v}}{\partial z}$   (6b)
$\overline{w'\theta'} = -K_h\frac{\partial \bar{\theta}}{\partial z}$   (6c)
The TDC is proportional to the square root of TKE, and can be expressed as:
$K_m = S_m l e^{1/2}$   (7a)
$K_h = S_h l e^{1/2}$   (7b)
where $l$ is mixing length and can be described as $l = \frac{l_0 \kappa z}{\kappa z + l_0}$, where $l_0 = \alpha \frac{\int_0^\infty z e^{1/2}dz}{\int_0^\infty e^{1/2}dz}$, $\alpha$ is an
empirical constant (=0.1). When $z$ converges to a very small value, $l$ converges to $\kappa z$. However, as
$z$ converges to a very large value, $l$ converges to $l_0$.
To obtain the $S_m$ and $S_h$ in Eq. (7), $G_m$ and $G_h$ are defined as:



**298** $\quad G_m = \frac{l^2}{2e}\left[\left(\frac{\partial \overline{u}}{\partial z}\right)^2 + \left(\frac{\partial \overline{v}}{\partial z}\right)^2\right]$ (8a)

**299** $\quad G_h = -\frac{l^2}{2e}\frac{g}{\theta_v}\frac{\partial \theta_v}{\partial z}$ (8b)

**300** $\quad$ $S_m$ and $S_h$ are functions of $G_m$ and $G_h$, and can be denoted as:

**301** $\quad S_m(6A_1 A_2 G_m) + S_h(1 - 3A_2 B_2 G_h - 12A_1 A_2 G_h) = A_2$ (9a)

**302** $\quad A_1(1 + 6A_1^2 G_m - 9A_1 A_2 G_h) - S_h(12A_1^2 G_h + 9A_1 A_2 G_h) = A_1(1 - 3C_1)$ (9b)

**303** $\quad$ where $[A_1, A_2, B_1, B_2, C_1] = [0.660, 0.657, 11.878, 7.227, 0.001]$.

**304** $\quad$ The PBLH in the MYJ scheme is defined as the height at which the TKE is reduced to a critical

**305** $\quad$ value of 0.1 m$^2$ s$^{-2}$.

**306** $\quad$ **2.3.3 ACM2 scheme**

**307** $\quad$ Unlike the YSU scheme, the ACM2 scheme applies the transilient matrix to deal with the

**308** $\quad$ contribution of nonlocal fluxes. The governing equation can be expressed as:

**309** $\quad \frac{\partial C_i}{\partial t} = f_{conv} M u C_1 - f_{conv} M d_i C_i + f_{conv} M d_{i+1} C_{i+1}\frac{\Delta z_{i+1}}{\Delta z_i} + \frac{\partial}{\partial z}\left[K_c(1 - f_{conv})\frac{\partial C_i}{\partial z}\right]$ (10)

**310** $\quad$ The first three terms on the right side of Eq. (10) represent nonlocal mixing effect and the fourth

**311** $\quad$ term represents local mixing effect. Where $C_i$ is the variable at layer $i$, $Mu$ is the nonlocal upward

**312** $\quad$ convective mixing rate, $Md_i$ is the downward mixing rate from layer $i$ to layer $i$-1, $\Delta z_i$ is the

**313** $\quad$ thickness of layer $i$, and $C_l$ represents the variable at the lowest layer in the model. $f_{conv}$ is the

**314** $\quad$ weighting factor for the nonlocal and local effects (i.e., $f_{conv} = \frac{K_h \gamma_h}{K_h \gamma_h - K_h \frac{\partial \theta}{\partial z}}$), where the value of $f_{conv}$

**315** $\quad$ ranges from 0 to 1, a larger $f_{conv}$ indicates stronger nonlocal mixing.

**316** $\quad$ There are two methods to calculate the TDC, and the first method is the same as the YSU scheme,

**317** $\quad$ i.e., Eq. (3), but there is also a very stable condition in the ACM2 scheme. In this case, the

**318** $\quad$ dimensionless function can be expressed as $\phi_m = \phi_h = \left(5 + \frac{0.1h}{L}\right)$.

**319** $\quad$ The second calculation principle is based on the mixing length theory, which uses mixing length

**320** $\quad$ and stability function to calculate TDC:

**321** $\quad K_h = 0.01 + l^2\sqrt{ss}f_h(Ri)$ (11)

**322** $\quad$ where $l$ is similar to the MYJ scheme, but $l_0$ is a constant (=80), $ss$ is the wind shear ($ss = $

**323** $\quad (\partial \overline{u}/\partial z)^2 + (\partial \overline{v}/\partial z)^2)$), 0.01 denotes the minimum value of the TDC in the model, and $f_h(Ri)$ is

**324** $\quad$ the empirical stability functions of gradient Richardson number of heat.

**325** $\quad$ i. $\quad$ when $Ri \geq 0$:

**326** $\quad f_h(Ri) = (1 - 25Ri)^{1/2}$ (12a)

**327** $\quad$ ii. $\quad$ when $Ri < 0$:

**328** $\quad f_h(Ri) = \frac{1}{1 + 10Ri + 50Ri^2 + 5000Ri^4} + 0.0012$ (12b)



Similarly, the empirical stability functions of momentum can be indicated as:
i.        when $Ri < 0$:
$f_m(Ri) = Pr \cdot f_h(Ri) + 0.00104$  (13a)
$K_m = 0.01 + l^2\sqrt{ss}f_m(Ri)$  (13b)
ii.        when $Ri \geq 0$:
$K_m = Pr \cdot K_h$  (13c)
The ACM2 scheme has a range setting for the TDC in the model with a minimum value of 0.01 m$^2$
s$^{-2}$ and a maximum value that cannot exceed 1000 m$^2$ s$^{-2}$.
The PBLH discrimination in the ACM2 scheme is similar to the YSU scheme, and is defined with
the bulk Richardson number method. The difference is that the entrainment region at the top of the
PBL is considered in the ACM2 scheme, and turbulence still exists due to the wind shear and thermal
penetration. Therefore, special processing of the PBLH is required under unstable and stable
conditions.
**2.3.4 BL scheme**
The BL scheme is also a one-and-a-half order local closure scheme, and the TDC is calculated in a
similar way to Eq. (7) of the MYJ scheme. Nevertheless, the function $S_m$ and mixing length (i.e., $l$)
are different from the MYJ scheme. In the BL scheme, $S_m$ is a constant 0.4 and the $l$ is divided into
upward and downward mixing length (i.e., $l_{up}$ and $l_{down}$), which are defined as:
$\int_z^{z+l_{up}} \beta[\theta(z) - \theta(z')]\,dz' = e(z)$  (14a)
$\int_{z-l_{down}}^z \beta[\theta(z') - \theta(z)]\,dz' = e(z)$  (14b)
where, $\beta$ is the buoyancy coefficient and the $l$ is equal to the minimum of $l_{up}$ and $l_{down}$ (i.e., $l =$
$min(l_{up}, l_{down})$). It is worth noting that in the BL scheme the TDC of heat is equal to the TDC of
momentum (i.e., $K_h = K_m$). In addition, the PBLH of the BL scheme is defined as the height at which
the virtual potential temperature of a layer is greater than that of the first layer by 0.5 K.
To accommodate different methods of calculating PBLH for different schemes and to evaluate the
simulation performance of PBLH, two methods are employed to calculated PBLH using observed
data in this study, the first being the bulk Richardson number method, and the detailed calculation
principle is as follows (Miao et al., 2018):
$Ri(z) = \frac{(g/\theta_s)(\theta_z - \theta_s)(z - z_s)}{(u_z - u_s)^2 + (v_z - v_s)^2 + bu_*^2}$  (15)
here $z$ is the height, $g$ is the gravity, $\theta$ is the virtual potential temperature, $u$ and $v$ are the components
of the horizontal wind, b is a constant, and $u_*$ is the friction velocity. The subscript "s" indicates
the near-surface. Since the friction velocity is much smaller in magnitude than the wind shear, the
b is set to 0, ignoring the effect of surface friction (Vogelezang and Holtslag, 1996; Seidel et al.,
2012). The PBLH is estimated as the lowest layer height when Ri reaches a critical value of 0.25.





The second method adopts the same calculation method as the BL scheme, i.e., the virtual potential
temperature method:
$\Delta\theta_{v|PBLH} = \theta_{v1} + 0.5$  (16)
The PBLH is the height when the virtual potential temperature exceeds the virtual potential
temperature of the first level by 0.5 K.
**2.4 Evaluation of the model**
To evaluate the PBL schemes and the performance of the model for estimating meteorological
variables, the statistical parameters used in this statistical analysis are defined as follows(Emery et
al., 2017):
Index of agreement (IOA):
$IOA = 1 - \dfrac{\left[\sum_{i=1}^{n}|X_{sim,i}-X_{obs,i}|^2\right]}{\left[\sum_{i=1}^{n}\left(|X_{sim,i}-\overline{X_{obs}}|+|X_{obs,i}-\overline{X_{obs}}|\right)^2\right]}$  (17)
Mean bias (MB):
$MB = \dfrac{1}{n}\sum_{i=1}^{n}\left(X_{sim,i} - X_{obs,i}\right)$  (18)
Root mean square error (RMSE):
$RMSE = \sqrt{\dfrac{1}{n}\sum_{i=1}^{n}\left(X_{sim,i} - X_{obs,i}\right)^2}$  (19)
Normalized standard deviations (NSD):
$NSD = \dfrac{\sqrt{\frac{1}{n-1}\sum_{i=1}^{n}\left(X_{sim,i}-\overline{X_{sim}}\right)^2}}{\sqrt{\frac{1}{n-1}\sum_{i=1}^{n}\left(X_{obs,i}-\overline{X_{obs}}\right)^2}}$  (20)
Relative bias (RB):
$RB = \dfrac{\overline{X_{sim}}-\overline{X_{obs}}}{\overline{X_{obs}}} \times 100\%$  (21)
Where $X_{sim,i}$ and $X_{sim,i}$ represent the value of simulation and observation, respectively, *i* refers
to time and *n* is the total number of time series. $\overline{X_{sim}}$ and $\overline{X_{obs}}$ represent the average simulation
and observation.
The Taylor diagram is a compact tool that displays simultaneously the values of four statistical
parameters: IOA, NSD, RB, and RMSE. In particular, in these diagrams the perfect match of a
model with the observations would be the point with IOA=1, NSD=1, RB=0 and RMSE=0.
**3 Results and discussion**
In section 3.1, the mechanistic analysis of the PBL schemes for the simulation of near-surface
meteorological parameters, including 2-m temperature, 2-m relative humidity, 10-m wind speed and
direction. Section 3.2 gives an in-depth analysis of different schemes for PBL vertical structure. In
section 3.3, the PBLH was evaluated for different schemes. In section 3.4, the reason for the





differences in turbulent diffusion are interrogated from the calculation principle of the schemes.
Section 3.5 summarizes the performance and expressiveness of different PBL schemes in different
regions, and recommends the optimal choice of PBL scheme.

**3.1 surface meteorological variables**

**3.1.1 2-m temperature and relative humidity**

To better analyze the variation of the time series, we selected representative stations in different
regions. Figure 2 shows the diurnal variation of 2-m temperature (i.e., $T_2$) for four months (i.e.,
January, April, July and October 2016) at representative sites (indicated in the orange dots in Fig. 1)
in the five regions. The model basically captures the daily variation characteristics of $T_2$, but there
are significant differences between different regions and seasons. The simulated results for July are
closest to the observed values (Fig. 2 c1-c5), anywhere. Overall, the mean biases (MBs) of the
diurnal variation of $T_2$ predicted in July for the NCP, YRD, SB, PRD and NS regions are 0.61~1.19,
-0.02~-0.56, -0.32~-0.60, -0.38~-0.69, and 0.28~0.81 ℃, respectively. However, a smaller value of
the mean bias does not mean that the simulated value of the model is closer to the observed value.
For example, if one overestimation and the other underestimation occur during the day and night,
the average results will cancel each other out, resulting in a small mean bias. Accordingly, more
statistical parameters are needed to further evaluate the optimal scheme. In the other three months
(January, April, and October), the simulated results of $T_2$ are overestimated to varying degrees
during daytime in the YRD, SB and PRD regions, while in the NS region, $T_2$ are underestimated to
varying degrees (Fig. 2 a2-b5, d2-d5 and Table 1). In the NCP regions, $T_2$ presents underestimation
in January by the model with the YSU, ACM2, BL and MYJ schemes are -1.33, -1.21, -0.52 and -
1.18 ℃, respectively, while overestimation arises in the other three months (Table 1). In the five
regions, the simulation results of the nighttime $T_2$ outperform those of the daytime $T_2$ for almost
four months. At night, the MYJ scheme shows a significant underestimation of $T_2$ for all months in
five regions compared to the other three schemes (Fig. 2 and Table 1). The simulation results of Hu
et al. (2010) and Xie et al. (2012) have also obtained the lowest temperature for the MYJ scheme
during the nighttime.

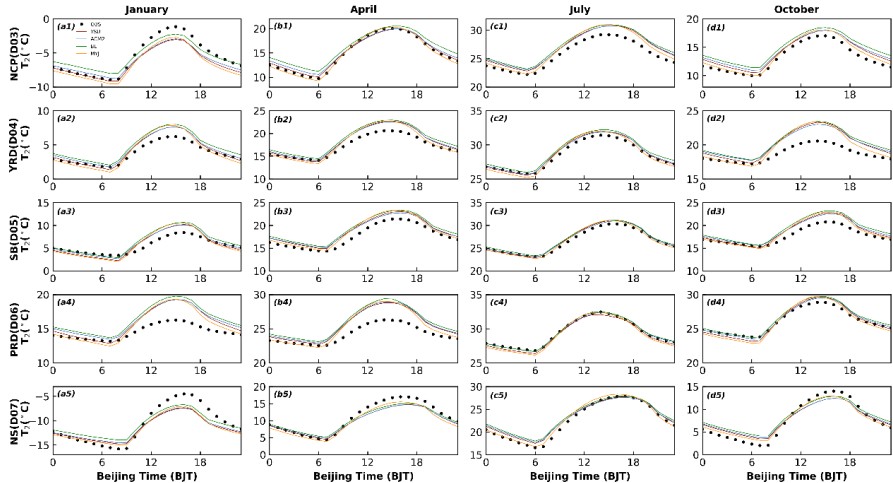

**420**

**421 Figure 2. Time series of diurnal variation of observed and simulated 2-m temperature in five**

**422 regions for four seasons.**

**423** The 2-m temperature does not actually represent the air temperature at a height of 2 m, but it is a

**424** diagnostic variable of the near-surface temperature. It is calculated from the surface temperature

**425** ($T_s$), the sensible heat flux (HFX) and the heat transfer coefficient ($C_h$). The $T_s$ is a prognostic

**426** variable, which is obtained in the model through the energy balance equation:

**427** $(1-\alpha)S\downarrow +L\downarrow -L\uparrow +G-HFX-LH=0$   (22)

**428** where $\alpha$ is the albedo of the underlying surface, $S\downarrow$ represents the downward of the shortwave

**429** radiation, $L\downarrow$ is the downward of the longwave radiation emitted by the cloud and atmosphere,

**430** $L\uparrow$ is the upward of the longwave emitted by the ground surface, G is the ground heat flux, and it

**431** is positive when heat transfers from the soil to the near surface, HFX is the sensible heat flux and

**432** LH is the latent heat flux.

**433** We compare the effects of the six variables mentioned above on $T_s$ with the expectation that we can

**434** further examine the reasons for the differences in $T_2$ variation between different schemes. The YSU

**435** scheme is used as a control and analyzed in comparison with each of the schemes.

**436** The nonlocal closure scheme (YSU) and the local closure scheme (MYJ) are compared first.

**437** Theoretically, the greater the downward shortwave radiation ($S\downarrow$) becomes, the more energy reaches

**438** the ground, and the higher the surface temperature ($T_s$) is. After comparing the YSU and MYJ

**439** schemes, the surface temperature does not show a proportional change with the downward

**440** shortwave radiation, and the $S\downarrow$ of the MYJ scheme is almost the same as that of the YSU scheme

**441** (Fig. 3 a1-e1), but the $T_s$ of the MYJ scheme is the lowest (Fig. 4 a1-e1). Therefore, the $S\downarrow$ is not

**442** the main factor that causes the difference in $T_s$ between the two schemes. There is no significant

**443** difference in the upward/downward longwave radiation between these two schemes (Fig. 3 a2-e3),

**444** so the effect of longwave radiation on the $T_s$ can also be excluded. During the daytime, the MYJ



scheme transfers less heat from the surface to the soil than the YSU scheme (Fig. 3 a4-e4), and the
$T_s$ of the MYJ scheme should be higher than that of the YSU scheme. But that's not how it has
turned out (Fig. 4 a1-e1). Thus, the ground heat flux (G) is also not a key factor that directly affects
the $T_s$. The latent heat flux (LH) is mainly related to water vapor (or relative humidity), so further
attention is paid to the effect of sensible heat flux (HFX) on $T_s$ (Fig. 3 a5-e6). The HFX is determined
by the difference between the surface temperature and the 2-m temperature ($T_s$-$T_2$), and the heat
transfer coefficient ($C_h$). MYJ has the largest HFX, and transfers more heat from the surface to the
atmosphere, resulting in the largest energy loss at the surface, which should correspond to the
smallest $T_s$ (Fig. 3 a6-e6, 4 a1-e1). The smallest difference between the two temperatures indicates
a smaller temperature gradient and more uniform mixing, symbolizing the largest $C_h$, which is also
true (Fig. 4 a2-e3). A larger $C_h$ would lead to higher $T_2$ during the day. Although the $T_s$ of the MYJ
scheme is significantly lower than YSU scheme, it makes the $T_2$ higher due to the large $C_h$. During
the daytime, the less heat is transferred from the surface to the soil in the MYJ scheme, which results
in lower soil temperature. During the nighttime, the difference in HFX and temperature gradient
between the two schemes decreases, and the lower soil temperature results in lower $T_s$ and $T_2$.

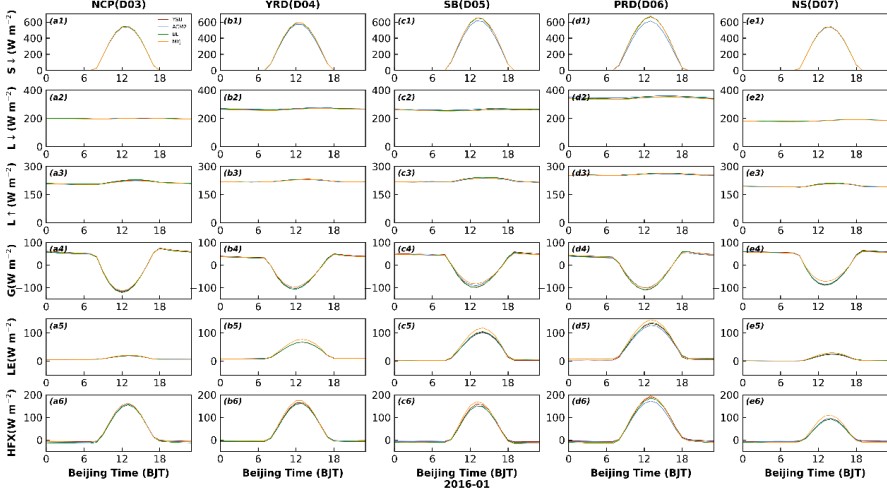


**Figure 3. Time series of diurnal variation of (a1-e1) downward shortwave radiation (S ↓), (a2-e2)**
**downward longwave radiation (L ↓), (a3-e3) upward longwave radiation (L ↑), (a4-e4) ground heat**
**flux (G), (a5-e5) latent heat flux (LH), and (a6-e6) sensible heat flux (HFX) by four PBL schemes**
**in five regions in January.**
The differences between the YSU scheme and the ACM2 scheme are further explored. Except for
the NCP and NS regions, the S ↓ of the ACM2 scheme is smaller than that of the YSU scheme in
the other three regions (i.e., YRD, SB and PRD) (Fig. 3 b1-d1). The HFX of the ACM2 scheme is
smaller than that of the YSU scheme (Fig. 3 b6-d6), the heat loss from the surface of the ACM2



scheme is less, and the $T_s$ of the ACM2 scheme should be higher. However, the $T_s$ corresponding to
the ACM2 scheme is lower than that of the YSU scheme (Fig. 4 b1-d1), reflecting that the $S\downarrow$
varies proportionally with the $T_s$, and it is the main factor controlling the $T_s$ variation. In the ideal
case, assuming the same temperature gradient for the nonlocal schemes, the $T_2$ of the YSU scheme
should also be higher than that of the ACM2 scheme when the $T_s$ of the YSU scheme is higher than
that of the ACM2 scheme with the same $C_h$. But in fact, it can be seen that the $C_h$ of the ACM2
scheme and YSU scheme are the same (Fig. 4 b2-d2), and the temperature gradient of the YSU
scheme is greater than that of ACM2 scheme (Fig. 4 b3-d3). The $T_2$ of the ACM2 scheme should be
slightly higher than the ideal case, closer to the $T_2$ of the YSU scheme, and even may also exceed
$T_2$ of the YSU scheme. At night, the $T_s$ of the YSU scheme is lower than that of the ACM2 scheme,
and the $C_h$ of the YSU scheme is smaller than that of the ACM2 scheme (Fig. 4 a1-e2). Meanwhile,
the difference in HFX between the two schemes is not obvious at night, contributing to lower $T_2$ of
the YSU scheme. In both NCP and NS regions, there is no significant difference in downward
shortwave radiation between two schemes, and no noticeable difference between $T_2$ and $T_s$.

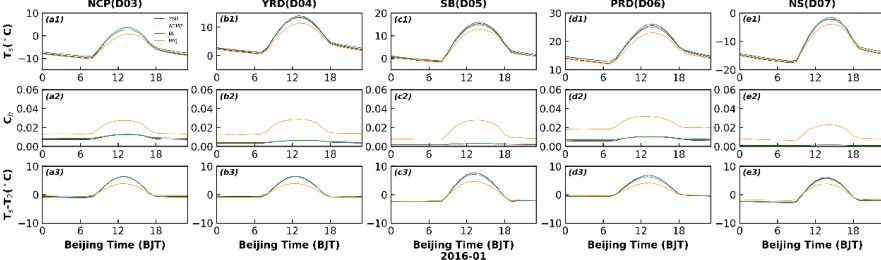


**Figure 4. Time series of diurnal variation of (a1-e1) surface temperature ($T_s$), (a2-e2) heat transfer**
**coefficient ($C_h$) and (a3-e3) the difference between the surface temperature and the 2-m**
**temperature ($T_s$-$T_2$) by four PBL schemes in five regions in January (Winter).**
Then, the reasons for the simulated temperature difference between the YSU scheme and the BL
scheme are demonstrated. During the daytime, the $S\downarrow$ of both schemes are the same (Fig. 3 a1-e1),
but the HFX of the BL scheme is smaller than that of the YSU scheme (Fig. 3 a6-e6), less heat is
loss at the surface, hence the $T_s$ should be higher than that of the YSU scheme (Fig. 4 a1-e1). The
$C_h$ of both schemes are the same, thus, the BL scheme has a higher $T_2$ (Fig. 2, 4a1-e2). At night, the
HFX of BL scheme is larger than that of YSU scheme, and more heat is transferred from atmosphere
to the surface, and the larger $C_h$ resulting in higher $T_2$ (Fig. 3 a6-e6, 4 a1-e2).
Finally, we can also uncover the reasons for the difference between the local closure schemes (MYJ
and BL). The larger HFX of the MYJ scheme leads to a lower $T_s$ in the daytime, while the
temperature gradient of MYJ scheme is smaller than that of the BL scheme, and $C_h$ is larger than
BL scheme (Fig. 3 a6-e6, 4). Therefore, the difference in $T_2$ between the two schemes is smaller
than that in $T_s$. The $T_2$ of the MYJ scheme is closer to that of the BL scheme.





In conclusion, the causes of temperature differences simulated by the nonlocal closure schemes
should first focus on the effect of the downward shortwave radiation ($S\downarrow$), and when it comes to the
local closure scheme, the effect of HFX should be further concerned. All of the above results have
been analyzed for January 2016, and the results for the other three months are similar (Figs. S1-S6).
In terms of regional distribution differences, $T_2$ in the northern and near mountainous regions of the
NCP region is significantly underestimated in the daytime for January, April and October, while $T_2$
in other regions shows an overestimation (Fig. 5, S7, S8 a1-e1). The range of overestimated areas
is smaller than the underestimated in January, only in a small part of the area south of Hebei Province
(Fig. 5 a1-e1). The relative bias (RB) of the underestimated (overestimated) $T_2$ with the YSU, ACM2,
BL and MYJ schemes are -0.60% (0.15%), -0.57% (0.17%), -0.43% (0.26%) and -0.60% (0.20%),
respectively in January. Also in these three months, temperature is overestimated at almost all
stations in the YRD region (RB=0.38%~0.50% in January, RB=0.49%~0.65% in April and
RB=0.58%~0.70% in October) and underestimated at some stations along the coast (RB=-0.13%~-
0.24% in January, RB=-0.32%~-0.37% in April and RB=-0.23%~-0.28% in October) (Fig. 5, S7,
S8 a2-e2). The results show an overestimation of $T_2$ simulated in those stations in the basin for the
SB region as well as the simulation results of the stations in the plain for the NCP region, while for
the stations in the hilltop areas, the $T_2$ shows an underestimation (Fig. 5, S7, S8 a3-e3). In the PRD
region, the entire region exhibits an overestimation of $T_2$, with the simulation results in October
(RB=0.06%~0.15%) being significantly better than those in January (RB=0.59%~0.81%) and April
(RB=0.56%~0.67%), with a lower degree of $T_2$ overestimation (Fig. 5, S7, S8 a4-e4). The BL
scheme simulates a higher $T_2$ and a large range of overestimated areas (about 167, 378, 252 and 100
stations in NCP, YRD, SB and PRD regions). The NS region has a more complex topography and
higher elevation, and the $T_2$ is underestimated at almost all stations, with best simulation results in
October (RB=-0.04%~-0.21%) and worst in April (RB=-0.49%~-0.64%).
For July, when the temperature is higher, the simulation results are significantly different from the
other three months. The relative bias of $T_2$ simulated with the YSU, ACM2, BL and MYJ schemes
are 0.47%, 0.46%, 0.53% and 0.46%, respectively, in the NCP region. The overestimation results
are similar to daytime, with the most pronounced overestimation for the BL scheme. For the
southern region of the NCP, the $T_2$ is consistently overestimated regardless of the season (Fig. S9
a1-e1). The $T_2$ at most stations are underestimated in the YRD region, which is different form the
other three months (Fig. S9 a2-e2). In summer, the temperature of the ocean, affected by the
subtropical high (prevailing southeasterly winds), is lower than that of the land, and the transport of
momentum is accompanied by the transport of heat from the sea to the land, causing the temperature
of the land to decrease. The $T_2$ of the basin area in the SB region is well reproduced, and no
significant overestimation occurs (Fig. S9 a3-e3). There is an underestimation of the $T_2$ at most
stations in the PRD region, but to a lesser extent (Fig. S9 a4-e4). In contrast, for the NS region, the



temperature is overestimated for areas at lower elevations, while underestimated (or better
reproduced) for areas at higher elevations (Fig. S9 a5-e5).

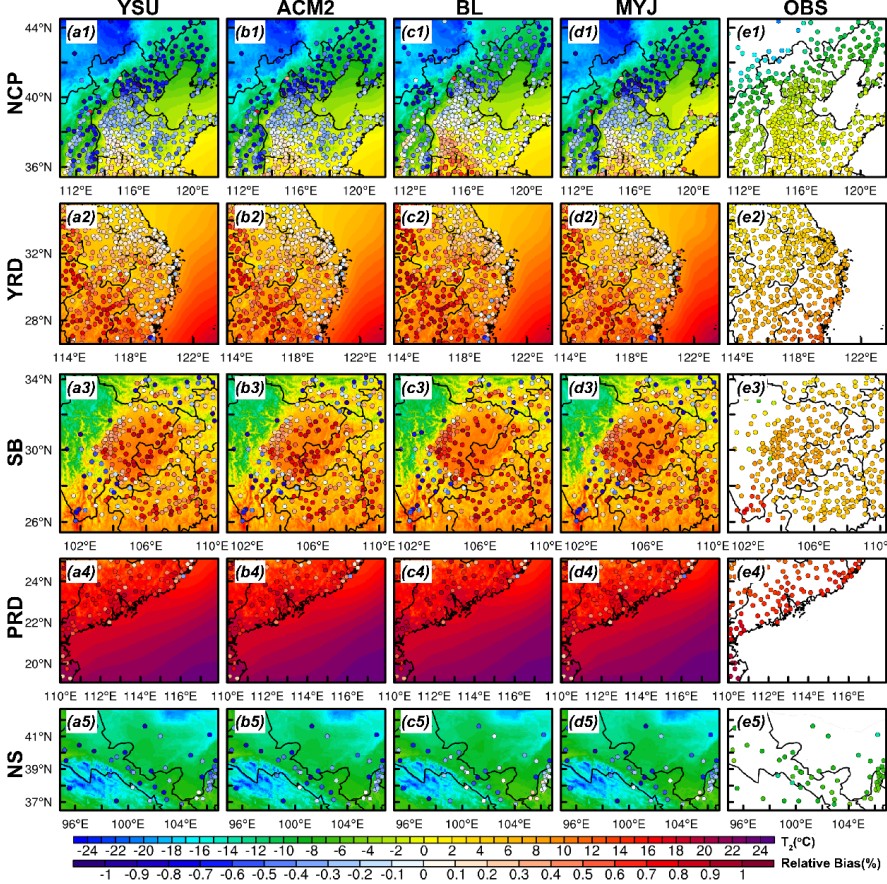


**Figure 5. Regional distribution of 2-m temperature simulated by (a-d) four PBL schemes in five**

**regions during the daytime in January (Winter), (e1-e5) distribution of observation in five regions,**

**and (a1-d5) distribution of relative bias between simulations and observations is denoted by**

**scatters.**

The relative deviation of the nighttime $T_2$ simulations is less than that of the daytime, regardless of
the region and month (Fig. 6, S10-S12). The differences between the four schemes are more striking
at night compared to the daytime. The BL scheme simulates the highest $T_2$ and the MYJ scheme
simulates the lowest $T_2$ in the whole region (Fig. 6, S10-S12). Compared to the observed values, the
MYJ scheme is the best when all schemes overestimate the simulated temperature, but if there is an
underestimation, the MYJ scheme is no longer the best scheme. Later, a comprehensive statistical
evaluation of the schemes will be presented. For the NCP region, the overestimation and
underestimation in the whole region do not show a north-south divide (or a mountain-plain divide)





**550** as in the daytime (Fig. 5, 6 a1-e1). In the YRD region, the temperature along the coastal area still

**551** shows a significant underestimation (Fig. 6, S10-S12 a2-e2). Similar to the daytime, the temperature

**552** at stations in the hill top areas of the SB region still presents an underestimation (Fig. 6, S10-S12

**553** a3-e3). Most stations show the underestimation of $T_2$ in July and October in the PRD region (Fig.

**554** S11-S12 a4-e4). In the NS region, the relative deviation of temperature simulations in October is

**555** greater than that in daytime (RB=0.17%~0.56%) (Fig. 5, S12 a5-e5).

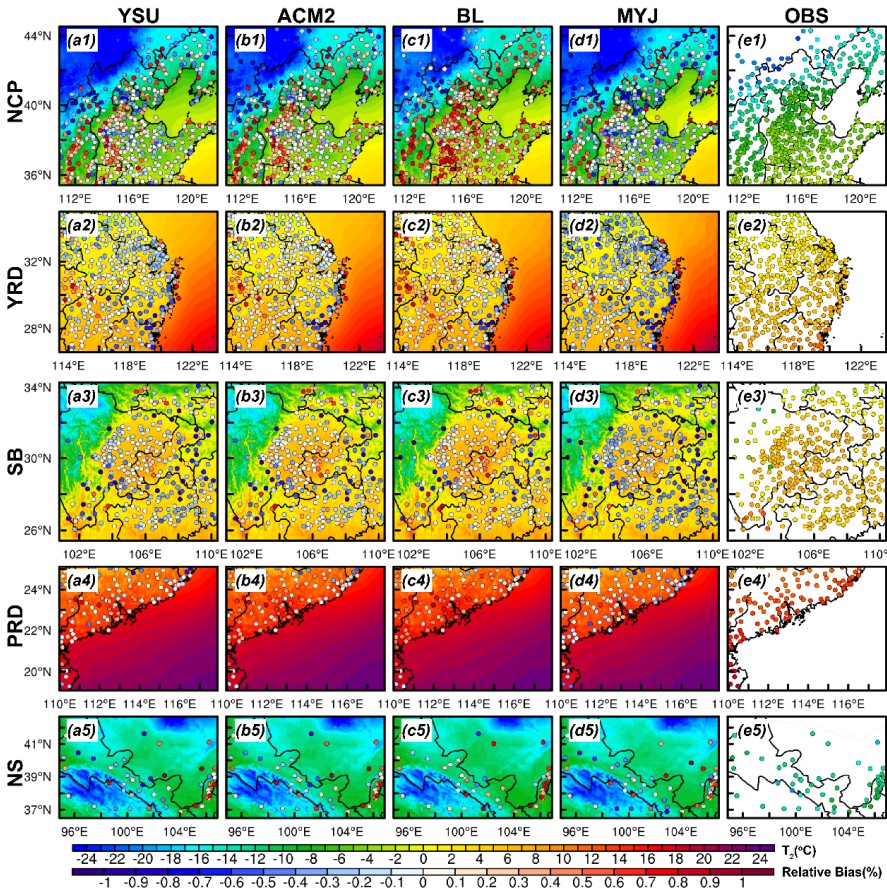

**556**

**557** **Figure 6. Similar as Figure 5, but at night.**

**558** In summary, the simulation results of $T_2$ have the following main characteristics. From the

**559** perspective of differences between observations and simulations, (1) the simulation results for July

**560** are better compared to the other three months. (2) The simulation results at night are better than

**561** those at daytime, with less relative deviation. (3) The temperature is easily underestimated at higher

**562** altitudes while overestimated in plains and basin areas. From the perspective of the differences

**563** between the different schemes, (1) the differences in the performance of the four schemes are more

**564** noticeable at night. (2) The difference in the simulation of temperature in the nonlocal closure



schemes is mainly attributed to the difference in downward shortwave radiation ($S\downarrow$), and the
difference in the variation of sensible heat flux (HFX) needs to be further analyzed when the local
closure schemes are involved. (3) The BL scheme simulates the highest temperature and the MYJ
scheme for the lowest temperature.
The results for 2-m relative humidity ($RH_2$) and $T_2$ correspond to each other, and the overestimation
of $T_2$ corresponds to the underestimation of $RH_2$. The simulation of $RH_2$ still shows the best results
in July, with the highest simulated values for the MYJ scheme and the lowest for the BL scheme.
Except for the NS region, the simulated $RH_2$ of the other four regions is almost underestimated.
This uniform trend in relative humidity may be due to errors in the initial field, which will be
discussed in Part II. Too much will not be repeated here (Figs. 2, 7).

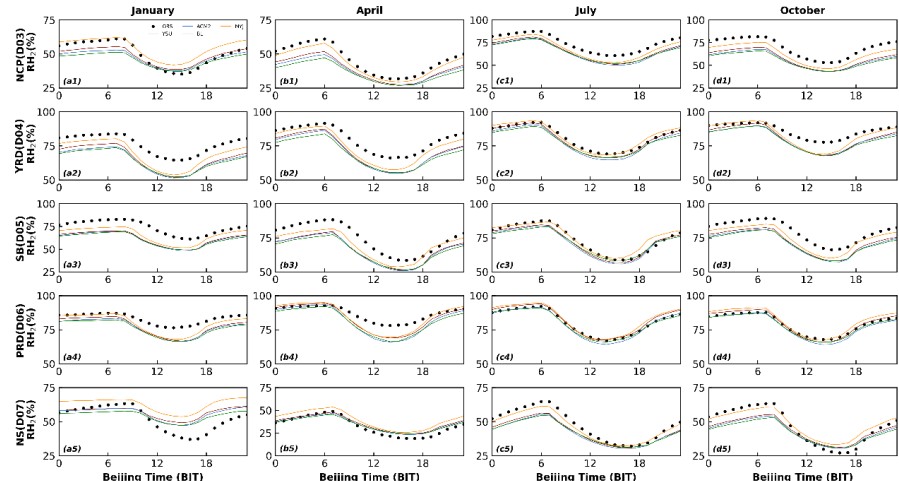

**Figure 7. Similar as Figure 2, but for 2-m relative humidity.**
**3.1.2 10-m wind speed and direction**
Although the model simulates the diurnal cycle of wind speed, the wind speed shows different
degrees of overestimation, and the mean bias is in the range of 0.86 m s$^{-1}$~2.74 m s$^{-1}$ (Fig. 8, Table
2). This is also the conclusion reached in many previous studies and it is more widely accepted by
the public. Except for the NCP region where the MYJ scheme has the largest mean bias (MB) value
during the day (YSU: MB=1.6 m s$^{-1}$, ACM2: MB=1.8 m s$^{-1}$, BL: MB=1.7 m s$^{-1}$, MYJ: MB=2.1 m
s$^{-1}$), while the BL scheme has the largest MB value at night regardless of the region (YSU: MB=1.3
m s$^{-1}$, ACM2: MB=1.8 m s$^{-1}$, BL: MB=2.2 m s$^{-1}$, MYJ: MB=1.7 m s$^{-1}$) (Fig. 8, Table 2).





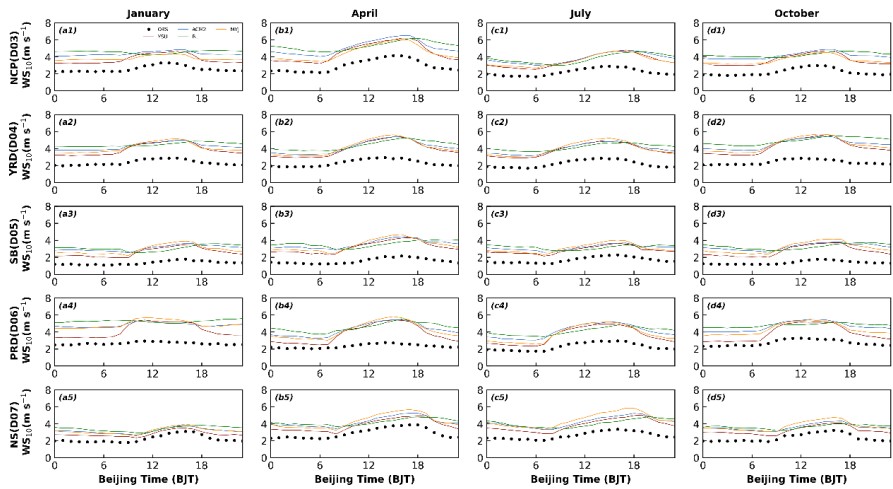

**Figure 8. Similar as Figure 2, but for 10-m wind speed.**

**Table 2. Mean bias of 2-m temperature, 2-m relative humidity and 10-m wind speed during daytime and nighttime by four PBL schemes in five regions and four seasons.**

| Variables | Schemes/Seasons | Regions | NCP | YRD | SB | PRD | NS |
|---|---|---|---|---|---|---|---|
| T$_2$-Day | YSU | Jan | -1.33 | 0.91 | 0.20 | 1.84 | -1.42 |
| | | Apr | 0.27 | 1.35 | 0.77 | 1.74 | -1.83 |
| | | Jul | 1.41 | -0.18 | -0.20 | -0.66 | 0.38 |
| | | Oct | 0.91 | 1.69 | 0.68 | 0.25 | -0.58 |
| | ACM2 | Jan | -1.21 | 0.97 | 0.29 | 1.93 | -1.29 |
| | | Apr | 0.34 | 1.28 | 0.64 | 1.79 | -1.82 |
| | | Jul | 1.37 | -0.21 | -0.31 | -0.49 | 0.27 |
| | | Oct | 0.98 | 1.54 | 0.61 | 0.21 | -0.50 |
| | BL | Jan | -0.52 | 1.33 | 0.73 | 2.32 | -0.76 |
| | | Apr | 0.89 | 1.71 | 1.07 | 2.00 | -1.43 |
| | | Jul | 1.60 | 0.04 | -0.16 | -0.38 | 0.55 |
| | | Oct | 1.49 | 1.88 | 1.09 | 0.46 | -0.09 |
| | MYJ | Jan | -1.18 | 0.92 | 0.32 | 1.70 | -1.23 |
| | | Apr | 0.45 | 1.19 | 0.87 | 1.66 | -1.39 |
| | | Jul | 1.38 | -0.29 | -0.28 | -0.58 | 0.64 |
| | | Oct | 0.83 | 1.56 | 0.68 | 0.18 | -0.45 |
| T$_2$-Night | YSU | Jan | -0.14 | -0.27 | -0.98 | 0.15 | 0.33 |
| | | Apr | 0.04 | -0.17 | 0.13 | 0.24 | -0.23 |
| | | Jul | 0.39 | -0.53 | -0.78 | -0.54 | 0.68 |
| | | Oct | 0.51 | 0.06 | -0.52 | -0.56 | 1.04 |
| | ACM2 | Jan | 0.15 | 0.03 | -0.74 | 0.56 | 0.52 |
| | | Apr | 0.47 | 0.07 | 0.23 | 0.39 | -0.07 |
| | | Jul | 0.57 | -0.40 | -0.72 | -0.27 | 0.86 |
| | | Oct | 0.86 | 0.35 | -0.29 | -0.28 | 1.24 |
| | BL | Jan | 0.88 | 0.34 | -0.45 | 0.80 | 1.03 |
| | | Apr | 1.21 | 0.48 | 0.53 | 0.64 | 0.23 |
| | | Jul | 0.79 | -0.08 | -0.48 | -0.21 | 1.07 |
| | | Oct | 1.37 | 0.54 | -0.06 | -0.04 | 1.59 |
| | MYJ | Jan | -0.46 | -0.59 | -1.16 | -0.28 | 0.21 |
| | | Apr | -0.70 | -0.62 | -0.27 | -0.19 | -0.97 |
| | | Jul | -0.15 | -0.83 | -0.93 | -0.80 | -0.08 |



| | | | | | | | |
|---|---|---|---|---|---|---|---|
| | | Oct | 0.01 | -0.34 | -0.74 | -0.83 | 0.52 |
| $RH_2$-Day | YSU | Jan | -0.01 | -9.95 | -10.71 | -6.62 | 7.19 |
| | | Apr | -7.09 | -7.59 | -5.75 | -5.02 | 3.02 |
| | | Jul | -8.46 | 1.18 | -0.16 | 3.18 | -5.56 |
| | | Oct | -11.07 | -5.62 | -5.17 | 0.57 | -1.02 |
| | ACM2 | Jan | -0.92 | -10.77 | -11.63 | -7.43 | 6.05 |
| | | Apr | -7.82 | -8.32 | -6.64 | -7.12 | 2.03 |
| | | Jul | -9.79 | -0.65 | -1.71 | 0.51 | -6.18 |
| | | Oct | -11.83 | -5.93 | -6.23 | -1.44 | -2.09 |
| | BL | Jan | -1.66 | -10.53 | -11.36 | -7.54 | 5.40 |
| | | Apr | -7.89 | -8.32 | -5.90 | -6.31 | 2.36 |
| | | Jul | -8.02 | 0.68 | 0.92 | 1.71 | -5.06 |
| | | Oct | -12.47 | -6.10 | -5.67 | -0.38 | -2.10 |
| | MYJ | Jan | 4.47 | -6.72 | -7.69 | -4.74 | 12.55 |
| | | Apr | -4.39 | -4.26 | -3.08 | -3.59 | 5.59 |
| | | Jul | -5.59 | 3.85 | 3.31 | 4.22 | -3.01 |
| | | Oct | -7.44 | -3.55 | -2.61 | 2.47 | 2.72 |
| $RH_2$-Night | YSU | Jan | -5.11 | -7.90 | -10.62 | -3.29 | -0.03 |
| | | Apr | -9.65 | -5.01 | -6.63 | 0.47 | -0.14 |
| | | Jul | -5.50 | 1.09 | -1.33 | 2.48 | -6.95 |
| | | Oct | -12.34 | -1.44 | -4.83 | 1.62 | -7.79 |
| | ACM2 | Jan | -6.38 | -9.54 | -11.03 | -4.56 | -0.48 |
| | | Apr | -11.35 | -5.91 | -7.05 | -0.86 | -1.15 |
| | | Jul | -6.86 | -0.64 | -2.25 | 0.18 | -7.96 |
| | | Oct | -14.11 | -2.92 | -5.69 | -0.94 | -8.85 |
| | BL | Jan | -8.40 | -10.47 | -11.89 | -5.01 | -2.36 |
| | | Apr | -13.51 | -8.00 | -8.25 | -2.07 | -2.16 |
| | | Jul | -7.32 | -1.75 | -2.47 | -0.23 | -8.06 |
| | | Oct | -15.92 | -3.51 | -6.19 | -0.91 | -9.94 |
| | MYJ | Jan | 1.84 | -3.86 | -6.91 | -0.69 | 6.86 |
| | | Apr | -3.89 | -1.65 | -3.44 | 1.91 | 5.84 |
| | | Jul | -1.68 | 3.02 | 0.42 | 3.61 | -1.91 |
| | | Oct | -7.18 | 1.04 | -2.50 | 3.36 | -1.45 |
| $WS_{10}$-Day | YSU | Jan | 1.33 | 1.92 | 1.58 | 2.17 | 0.59 |
| | | Apr | 1.86 | 1.97 | 2.04 | 2.25 | 0.93 |
| | | Jul | 1.35 | 1.56 | 1.20 | 1.79 | 1.30 |
| | | Oct | 1.68 | 2.11 | 1.54 | 1.70 | 0.93 |
| | ACM2 | Jan | 1.57 | 2.04 | 1.79 | 2.26 | 0.91 |
| | | Apr | 2.11 | 2.01 | 2.18 | 2.37 | 1.21 |
| | | Jul | 1.43 | 1.62 | 1.30 | 2.02 | 1.50 |
| | | Oct | 1.90 | 2.19 | 1.73 | 2.05 | 1.21 |
| | BL | Jan | 1.50 | 2.02 | 1.63 | 2.40 | 1.01 |
| | | Apr | 1.85 | 2.04 | 1.93 | 2.44 | 0.86 |
| | | Jul | 1.21 | 1.54 | 1.12 | 1.72 | 1.10 |
| | | Oct | 1.83 | 2.28 | 1.60 | 1.95 | 1.04 |
| | MYJ | Jan | 1.63 | 2.26 | 2.14 | 2.67 | 1.10 |
| | | Apr | 2.33 | 2.40 | 2.65 | 2.69 | 1.61 |
| | | Jul | 1.85 | 2.05 | 1.85 | 2.16 | 2.09 |
| | | Oct | 2.01 | 2.58 | 2.17 | 2.12 | 1.55 |
| $WS_{10}$-Night | YSU | Jan | 1.26 | 1.43 | 1.50 | 1.40 | 0.88 |
| | | Apr | 1.51 | 1.49 | 1.67 | 1.22 | 1.07 |
| | | Jul | 1.16 | 1.32 | 1.15 | 1.16 | 1.07 |
| | | Oct | 1.42 | 1.45 | 1.40 | 1.14 | 1.04 |
| | ACM2 | Jan | 1.88 | 1.91 | 1.97 | 2.21 | 1.36 |
| | | Apr | 2.15 | 1.81 | 2.07 | 1.79 | 1.53 |
| | | Jul | 1.56 | 1.62 | 1.50 | 1.71 | 1.59 |
| | | Oct | 1.98 | 1.92 | 1.80 | 1.98 | 1.59 |



| | | | | | | |
|---|---|---|---|---|---|---|
| | Jan | 2.32 | 2.32 | 2.13 | 2.74 | 1.63 |
| | Apr | 2.72 | 2.28 | 2.38 | 2.33 | 1.73 |
| BL | Jul | 1.79 | 2.09 | 1.75 | 2.08 | 1.71 |
| | Oct | 2.38 | 2.44 | 2.06 | 2.38 | 1.78 |
| | Jan | 1.63 | 1.76 | 1.94 | 2.18 | 1.45 |
| | Apr | 1.79 | 1.70 | 2.12 | 1.71 | 1.53 |
| MYJ | Jul | 1.42 | 1.56 | 1.52 | 1.42 | 1.66 |
| | Oct | 1.74 | 1.81 | 1.83 | 1.68 | 1.65 |


Similar to $T_2$, 10-m wind speed (i.e., $WS_{10}$) is also a diagnostic variable of the near-surface wind
speed. For the YSU, ACM2 and BL schemes, in the revised MM5 surface layer scheme, $WS_{10}$ is
calculated based on the Monin-Obukhov (M-O) similarity theory(Monin and Obukhov, 1954). The
dimensionless profile function of momentum is denoted as:
$$\phi_m\left(\frac{z}{L}\right) = \frac{\kappa z}{u_*}\frac{\partial u}{\partial z} \quad (23)$$
where $\kappa$ is the von Karman constant, $u_*$ is the friction velocity, $z$ is the height, $L$ is the Obukhov
length, integrating the Eq. (23) with respect to height z:
$$du = \frac{u_*}{\kappa}\left[\frac{dz}{z} - \frac{1 - \phi_m\left(\frac{z}{L}\right)}{\frac{z}{L}}d\left(\frac{z}{L}\right)\right] \quad (24)$$
integrate Eq. (24):
$$\int_0^u du = \frac{u_*}{\kappa}\left\{\int_{z_0}^z \frac{dz}{z} - \int_{\frac{z_0}{L}}^{\frac{z}{L}}\left[1 - \phi_m\left(\frac{z}{L}\right)\right]d\ln\left(\frac{z}{L}\right)\right\} \quad (25)$$
here, let $\psi_m\left(\frac{z}{L}\right) = \int_0^{\frac{z}{L}}\left[1 - \phi_m\left(\frac{z}{L}\right)\right]d\ln\left(\frac{z}{L}\right)$, where $\psi_m\left(\frac{z}{L}\right)$ is the integrated similarity function
for momentum.
Therefore, Eq. (25) can be indicated as $u = \frac{u_*}{\kappa}\left[\ln\left(\frac{z}{z_0}\right) - \psi_m\left(\frac{z}{L}\right) + \psi_m\left(\frac{z_0}{L}\right)\right]$, where $z_0$ is the
roughness length.
Based on the bulk transfer method, the momentum flux can be represented as $\tau = \rho u_*^2 = \rho C_m u^2$,
where $\tau$ is the momentum flux, $C_m$ is the bulk transfer coefficient for momentum:
$$C_m = \frac{u_*^2}{u^2} = \frac{\kappa^2}{\left[\ln\left(\frac{z}{z_0}\right) - \psi_m\left(\frac{z}{L}\right) + \psi_m\left(\frac{z_0}{L}\right)\right]^2} \quad (26)$$
Thus, the wind speed at 10 m divided by the wind speed at a certain height can be written as:
$$u_{10} = \frac{u_*}{\kappa}\left[\ln\left(\frac{z}{z_0}\right) - \psi_m\left(\frac{z}{L}\right) + \psi_m\left(\frac{z_0}{L}\right)\right] \cdot \frac{\left[\ln\left(\frac{10}{z_0}\right) - \psi_m\left(\frac{10}{L}\right) + \psi_m\left(\frac{z_0}{L}\right)\right]}{\left[\ln\left(\frac{z}{z_0}\right) - \psi_m\left(\frac{z}{L}\right) + \psi_m\left(\frac{z_0}{L}\right)\right]} = \frac{u_*}{\kappa}\left[\ln\left(\frac{z}{z_0}\right) - \psi_m\left(\frac{z}{L}\right) + \right.$$
$$\left. \psi_m\left(\frac{z_0}{L}\right)\right] \cdot \left(\frac{C_m}{C_{m10}}\right)^{1/2} \quad (27)$$
where $C_{m10}$ is the transfer coefficient for momentum at 10 m height:



$$C_{m10} = \frac{\kappa^2}{\left[\ln\left(\frac{10}{z_0}\right) - \psi_m\left(\frac{10}{L}\right) + \psi_m\left(\frac{z_0}{L}\right)\right]^2} \quad (28)$$
Comparing the $C_m$ of the three schemes (i.e., YSU, ACM2 and BL schemes) at night, $C_m$ is the
largest for the BL scheme, the second largest for the ACM2 scheme, and the smallest for the YSU
scheme (Fig. 9). Correspondingly, the BL scheme simulates the largest WS10, ACM2 the second
largest, and the YSU the smallest (Fig. 8). The larger $C_m$ corresponds to the stronger mixing, which
transports more momentum from the upper to the lower layers, making $WS_{10}$ increase. Therefore,
the bulk transfer coefficient $C_m$ controls the variation of $WS_{10}$ at night. During the daytime, the $C_m$
of the BL scheme is smaller than that of the other two schemes, and the corresponding $WS_{10}$ decrease
(Fig. 8, 9). However, the difference among the three schemes is smaller in daytime than that in
nighttime. The reason why the results of $C_m$ and $WS_{10}$ differ with the same calculation method is
because of the vertical variation of heat and momentum within the boundary layer involved in the
calculation. This will correlate to the vertical diffusion coefficients within the boundary layer that
will be discussed further in a later section.

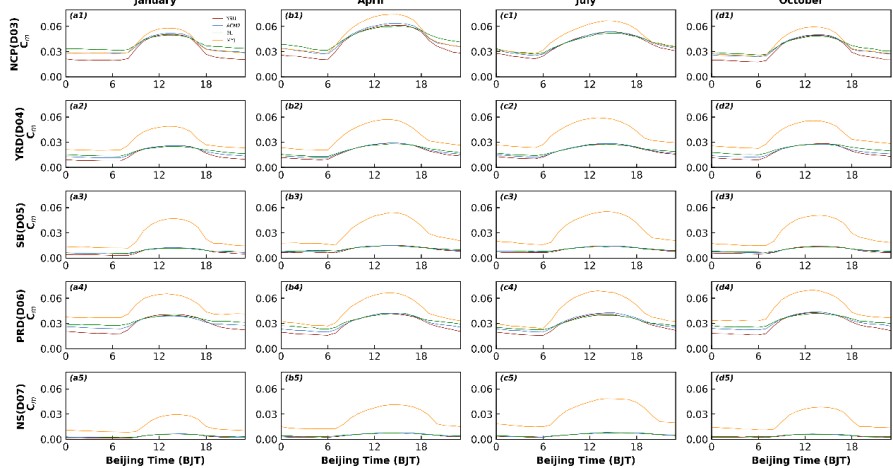


**Figure 9. Similar as Figure 2, but for momentum transfer coefficient ($C_m$).**
For the near surface scheme of the MYJ scheme, $WS_{10}$ is calculated according to the near surface
flux profile relationship proposed by Liu et al. (1979):
$$u_0 - u_s = D_1\left[1 - exp\left(-\frac{z_u u_*}{D_1 \nu}\right)\right]\left(\frac{F_u}{u_*}\right) \quad (29)$$
where 0 represents the value at height $z$ above the surface where the molecular diffusivity still plays
a dominant role, $s$ denotes the surface value, $D_1$ denotes a near surface parameter, $u_*$ is the friction
velocity, $\nu$ is the molecular diffusivity for momentum ($= 1 \times 10^{-5}$), and $F_u$ is the momentum flux.
Since $1 - exp\left(-\frac{z_u u_*}{D_1 \nu}\right) \approx \frac{z_u u_*}{D_1 \nu}$, $u_0 - u_s = \left(\frac{z_u}{\nu}\right) F_u$.



The momentum flux in the surface layer above the viscous sublayer is represented by $F_u =$
$\left(\frac{C_m}{\Delta z_e}\right)(u_{low} - u_0)$, here, the subscript *low* denotes the variables at the lowest model level, $\Delta z_e$ is
either the equivalent height of the lowest model level that considers the presence of the "dynamical
turbulence layer" at the bottom of the surface layer (Janjić, 1990). $C_m$ is the bulk transfer coefficient,
defined as:
$$C_m = \frac{\kappa u_*}{\ln\left(\frac{z_0+z}{z_0}\right) + \psi_m\left(\frac{z_0+z}{L}\right) - \psi_m\left(\frac{z_0}{L}\right)} \quad (30)$$
In Eq. (29), $z_u$ is still an unknown, such that $\frac{z_u u_*}{D_1 \nu} = \xi$, where $\xi$ is a smaller constant (equal to
0.35 in the model). Here, the near surface parameter $D_1$ is further defined as $D_1 = C \cdot$
$\left(\frac{z_0 u_*}{\nu}\right)^{1/4}$, where C is a constant (=30), the roughness length $z_0$ as a function of $u_*$ ($z_0 = \frac{0.11\nu}{u_*} +$
$\frac{0.018 u_*^2}{g}$), substituting $D_1$ and $z_u$ to Eq. (29):
$$u_0 = \frac{\frac{\xi}{u_*}\left[C \cdot \left(\frac{z_0 u_*}{\nu}\right)^{1/4}\right]\left(\frac{C_m}{\Delta z_e}\right)u_{low} + u_s}{1 + \frac{\xi}{u_*}\left[C \cdot \left(\frac{z_0 u_*}{\nu}\right)^{1/4}\right]\left(\frac{C_m}{\Delta z_e}\right)} \quad (31)$$
The wind speed at a height of 10 m can be expressed as:
$$u_{10} = \frac{F_u \Delta z_e}{C_{m10}} + u_0 = \frac{C_m(u_{low} - u_0)}{C_{m10}} + u_0 \quad (32)$$
where $C_{m10}$ is the transfer coefficient for momentum at 10 m height:
$$C_{m10} = \frac{\kappa u_*}{\ln\left(\frac{z_0+10}{z_0}\right) + \psi_m\left(\frac{z_0+10}{L}\right) - \psi_m\left(\frac{z_0}{L}\right)} \quad (33)$$
Therefore, the $C_m$ of the MYJ scheme is significantly different from the other three schemes. Except
for the NCP region, although the $C_m$ of the MYJ scheme is larger than the other three schemes at all
times of the day, the $WS_{10}$ presents the maximum only during the daytime. This suggests that at
night, the wind speed simulated by the MYJ scheme also be influenced by other factors. For example,
the calculation method of the integrated similarity functions ($\psi_m$) in the MYJ scheme is different
from the other three schemes. In the other three schemes, the $\psi_m$ is calculated according to four
stability regimes defined in terms of the bulk Richardson number(Zhang and Anthes, 1982). In the
MYJ scheme, the $\psi_m$ is calculated based on two stability regimes by the z/L (Paulson, 1970).



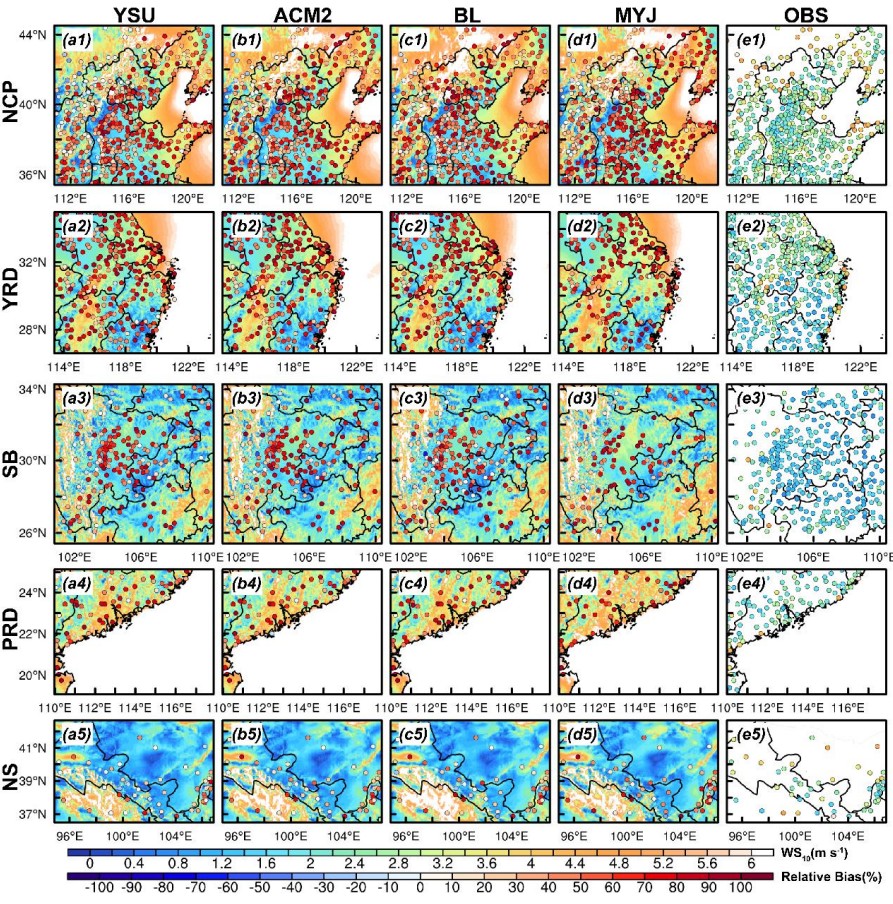

**Figure 10. Similar as Figure 5, but for 10-m wind speed.**

The reasons for the differences in WS$_{10}$ simulation are further analyzed in terms of regional distribution. During the daytime, wind speed is significantly overestimated at most sites throughout the NCP region, which are centered in the plains and valleys, but is less overestimated and even underestimated at some sites on the mountain tops (Fig. 10, S13-S15 a1-e1). Wind speed is overestimated at almost all stations throughout the YRD and PRD regions (Fig. 10, S13-S15 a2-e2, a4-e4). The WS$_{10}$ in the basin is importantly overestimated in the SB region, while less overestimated at hilltop stations on the eastern side of the basin, with higher wind speed being more pronounced in January (Fig. 10, S13-S15 a3-e3). In the NS region, wind speed is overestimated to a lesser extent than in other regions, but for regions with lower wind speed, the relative bias (RB) is larger, especially in July (Jan: RB=29.3%~49.1%, Apr: RB=32.0%~55.6%, Jul: RB=44.2%~78.7%, Oct: RB=42.7%~65.7%). Comparing the simulation results of the four schemes, the MYJ scheme simulates the most significantly overestimated wind speed and the least overestimated for the YSU scheme (Fig. 10, S13-S15). In comparison with the four months, it is



found that the RB of the simulation is the largest for the month with slower wind speed (i.e., July).
At night, the wind speed is overestimated at almost all stations in the whole region of NCP, and the
overestimation is greater at the hilltop stations than during the day (Fig. 11, S16-S18). The other
four regions are more similar to the daytime (Fig. 11, S16-S18). However, by comparing the four
schemes, we find that the BL scheme has the most obvious overestimation, different from the
daytime, while the YSU scheme still has the lowest overestimation, the same as the daytime. In
general, wind speed is smaller at night, and the four schemes overestimate wind speed much more
than during the day. Averaging the RB of wind speed over the five regions and four months, the
daytime (nighttime) values for the YSU, ACM2, BL and MYJ schemes are 77.7% (92.4%), 85.6%
(123.6%), 80.2% (146.0%), and 100.8% (117.4%), respectively. This simulated misestimation of
low winds at night may mainly originate from the inapplicability of the M-O similarity theory. The
strong stable boundary layer usually occurs on nights with low winds(Monahan and Abraham, 2019;
Vignon et al., 2017). In this strong stable boundary layer, turbulence occurs weakly and
intermittently, the turbulence intensity is disproportionate to the mean gradient, and the M-O
similarity theory is no longer applicable(Acevedo et al., 2015; Sun et al., 2012). Ultimately, these
inapplicable functions affect the calculation of the bulk transfer coefficient and can further lead to
large deviations in the simulation of wind speed.

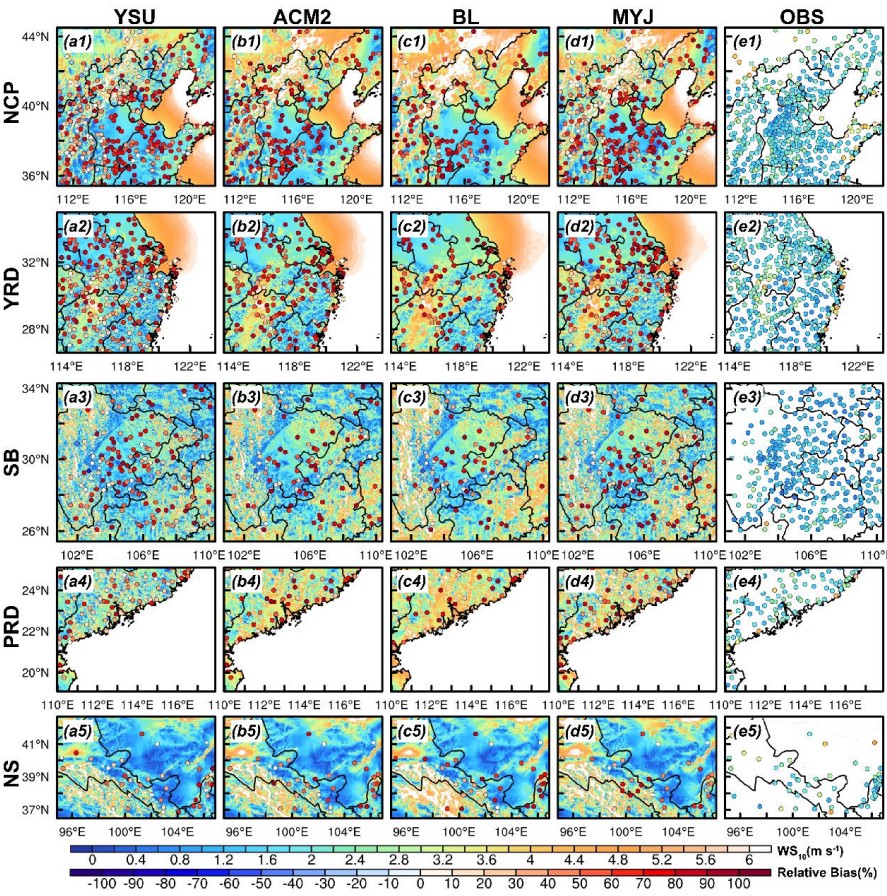

**Figure 11. Similar as Figure 6, but for 10-m wind speed.**

We further re-analyze the effect of topography on wind speed. The wind speed is overestimated for plains and valleys and better reproduced/underestimated for mountain tops, mainly because of the smoother topography in the model. This is rather because coastal stations in the plains, many of which also have high wind speeds, are not well reproduced and still show significant overestimation (Fig. 10), than the high wind speeds at the top of the mountains, which are better simulated. It is assumed that the wind speed should be small in plain areas with complex underlying surface, but it increases after the model has smoothed the terrain. The wind speed increases gradually with height, and when the terrain at the top of the mountain is smoothed, the originally larger wind speed decreases, and the wind speed will be closer to the observed value.



**699**

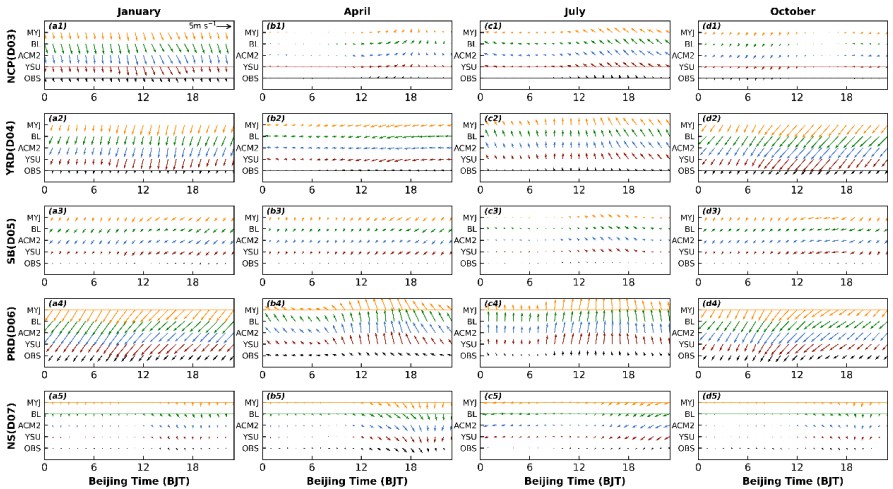

**700**  **Figure 12. Similar as Figure 2, but for 10-m wind direction.**

**701**  The model can basically simulate the changes of wind direction in the five regions, well capturing

**702**  the overall wind direction in each region (Fig. 12). In the NCP region, the simulation of wind

**703**  direction is poor in January compared to the other three months, with a high frequency of

**704**  northwesterly-northerly winds, overestimated by about 6.6% (Fig. 13 a1-d1). In addition, coupled

**705**  with larger wind speed, it causes the effect of advective transport to be amplified, thus affecting the

**706**  variation of pollutant concentrations(W. Jia and Zhang, 2021). The frequency of simulated

**707**  northeasterly winds in the YRD region is higher than that observed in January (~6.9%), April

**708**  (~6.9%) and October (~11.2%), while the frequency of southerly winds is higher in July (~10.0%)

**709**  (Fig. 13 a2-d2). The wind direction of SB region is poorly simulated since the topography is too

**710**  complicated in the SB region, with a basin in the middle and high topographic mountains all around

**711**  (Fig. 13 a3-d3). The low wind state in the middle of the basin is difficult to be captured. The

**712**  percentage of northeasterly winds simulated by the model in January, April, July and October are

**713**  22.9%~25.6%, 19.2%~20.6%, 14.9%~16.4%, 22.4%~24.2%, respectively, and the percentage of

**714**  observations are 15.1%, 12.0%, 10.9%, 16.1%, respectively. Similarly, the percentage of westerly

**715**  winds simulated by the model in January, April, July and October are 4.7%~5.1%, 4.7%~5.7%,

**716**  4.8%~5.5%, 3.8%~5.1%, respectively, and the percentages of observations are 9.9%, 12.4%, 12.4%,

**717**  12.3%, respectively. The model simulates a large proportion of northeasterly winds and a smaller

**718**  proportion of westerly winds (Fig. 13 a3-d3). In the PRD region, the frequency of northeasterly

**719**  wind occurrences in January and October is significantly overestimated by about 8.8% and 9.5%,

**720**  while the frequency of southerly winds is overestimated in April and July by about 14.5% and 17.3%,

**721**  and the frequency of southeasterly winds is underestimated (Fig. 13 a4-d4). The wind direction is

**722**  better simulated in the NS region, not significantly influenced by the complex terrain (Fig. 13 a5-



723   d5).

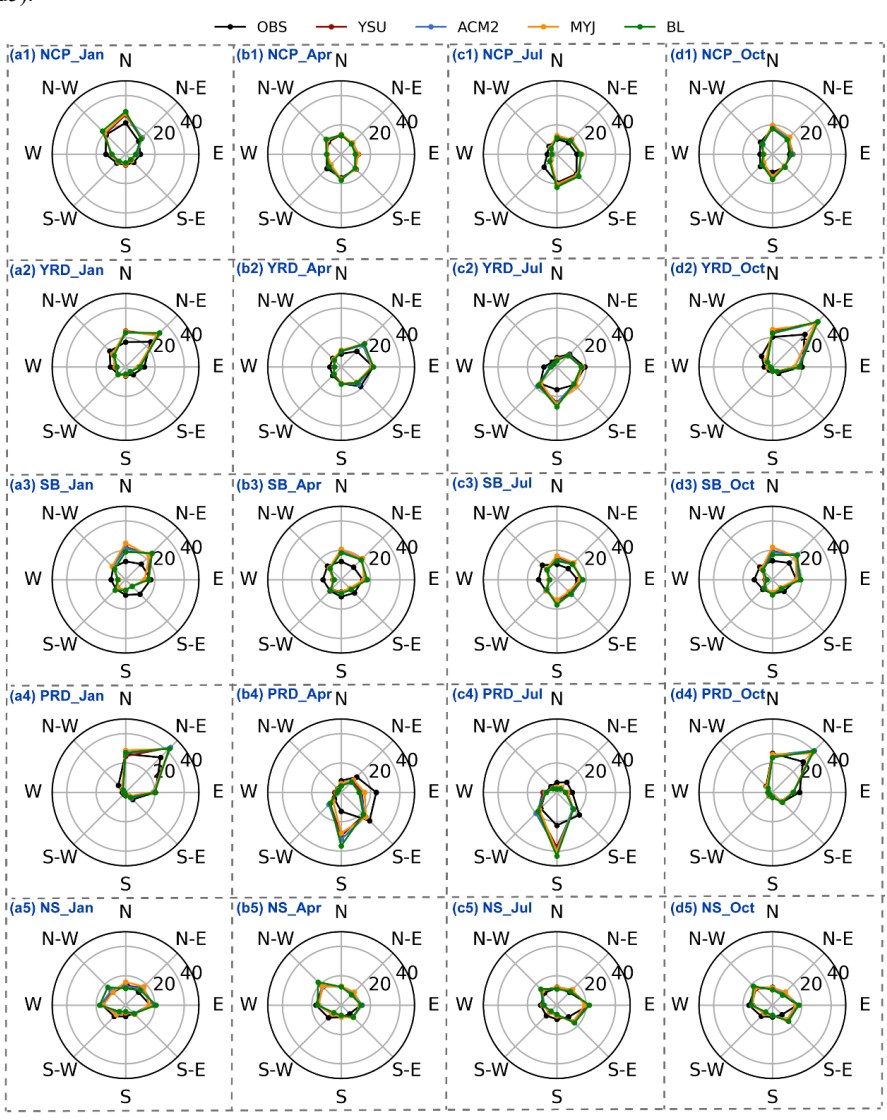

**Figure 13.** Wind-rose plots in five regions for four seasons are (a1-d1) NCP region, (a2-d2) YRD region, (a3-d3) SB region, (a4-d4) PRD region and (a5-d5) NS region, respectively.

### 3.2 Vertical structures

To better understand the performance of model in simulating PBL structure under different underlying surface, four representative stations have been selected in each region, with stations in plain areas, stations in mountains areas with high elevation, and stations near the sea.





### 3.2.1 temperature

Accurate simulation of the vertical structure of the PBL is very important for the evolution of pollution, precipitation and typhoons. In the vertical direction, four typical sounding stations are selected for each region at 08:00 to better reflect the simulation of the vertical structure of the PBL under different underlying surface conditions. Overall, the model captures the vertical structures of the temperature. From the simulation results in January, the best reproduction of the temperature simulation is found in the YRD region, in which the temperature is closer to the observed values (Fig. 14 a2-d2). In addition to the NCP, SB and NS regions, a temperature inversion layer appears in the lower layers at 08:00, and the NS region has the most significant temperature inversion (Fig. 14 a1-d1, a3-d3, a5-d5). The model does not simulate the temperature variation of the inversion layer well, and shows significant differences from the observations. When there is a difference in topography between the observed and simulated stations, the bias in the temperature is more pronounced. These stations usually exist in complex topographic conditions, such as Zhangjiakou and Zhangqiu stations in the NCP region, Shapingba in the SB region, and Mazongshan and Jiuquan in the NS region (Fig. 14 b1, d1, c3, a5, b5). The topographic discrepancy caused by the lack of high resolution may, on the one hand, account for it, resulting in more complex topography in the grid points closest to the observation stations. On the other hand, there is also an urgent need for finer underlying surface data to respond more closely to the observed real topography. The effect of resolution and underlying surface will be discussed in detail in the **Part II**. Although the elevation of the Hongyuan station in the SB region is higher, the difference in topographic height obtained from observations and simulations are close to each other (Fig. 14 a3).

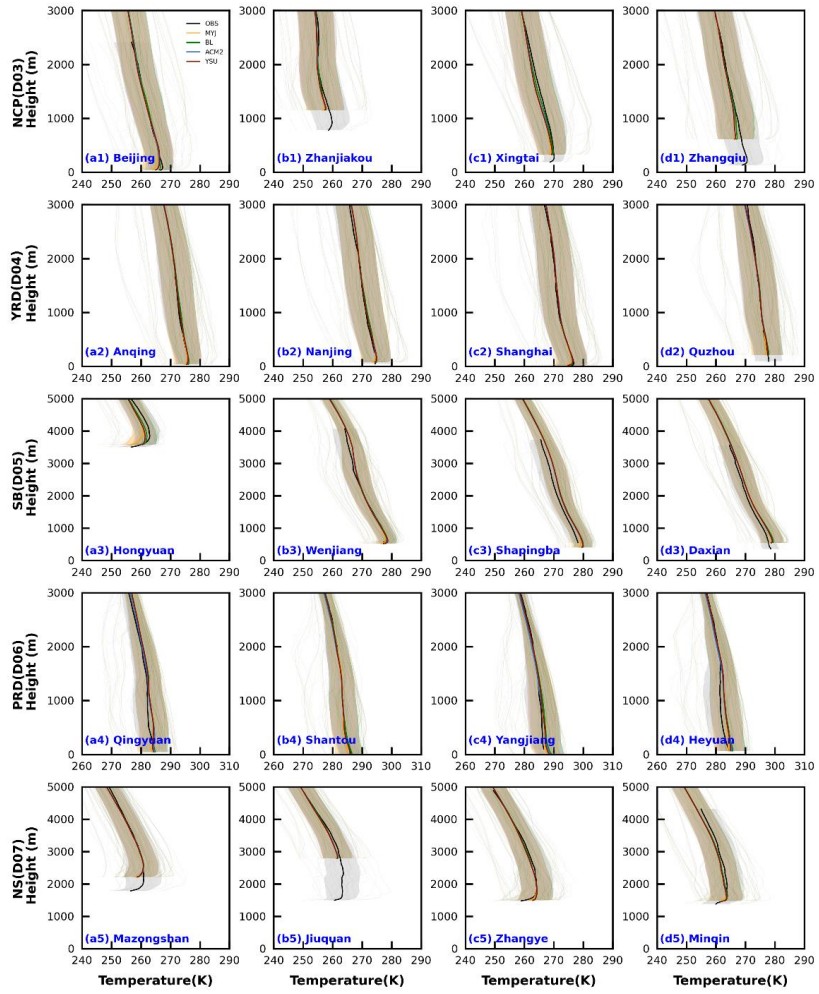

**Figure 14. Average vertical profiles of observed and simulated temperature at 08:00 and 20:00 BJT at four sounding stations for each region in January (Winter). The unobtrusive gray lines indicate the simulated lines for all time periods, and the lines with shading indicate the average values and shaded areas show the uncertainty range (the mean ±1 standard deviation).**

There is an underestimation of temperature at stations with higher topography and overestimation for lower topography, which is more consistent with the conclusions drawn from 2-m temperature (Fig. 5, 14). However, the underestimation of temperature is not present throughout the vertical, but is more pronounced in the lower layers, which are more influenced by the underlying surface. From the differences of the four schemes, the MYJ scheme simulates the lowest temperature and largest temperature gradient. Since the MYJ scheme simulates a weak turbulent diffusion of heat, a well vertical exchange process cannot occur, bringing into a large temperature gradient (Fig. S19). The



differences of the four schemes gradually decrease with the increase of the height. The BL scheme,
which is also a local closure scheme, with a smaller vertical gradient in temperature, mainly because
this scheme adds a counter-gradient correction term to the heat flux, which is mainly applicable to
the convective PBL(Bougeault and Lacarrere, 1989). The presence of this term leads to an increase
in turbulent diffusion and a decrease in temperature gradient. However, it is worth noting that there
are still slightly stable stratifications at 08:00, and this term generates upward heat flux and reduces
the temperature gradient, which is closer to the results of the nonlocal closure schemes (YSU and
ACM2 schemes). The simulation results for the other three months are not as good as January, but
the simulation characteristics are similar to January (Figures not shown). The results at 20:00 are
similar to those at 08:00, and thus will not be repeated here (Figures not shown).
**3.2.2 wind speed and direction**
The simulation of wind speed vertical structure is much worse in comparison to temperature (Fig.
14, 15). The simulated results of wind speed in the vertical direction and 10-m wind speed are still
quite different. In the four months, the wind speed is almost overestimated at the lower altitude
stations below 1000 m in all the four regions except the NS region, and wind speed is less
overestimated in July than in the other three months (Fig. 15, S20-S22 a1-d4). However, for the NS
region, the wind speed is almost better simulated, or underestimated, and is significantly different
from the other four regions (Fig. 15, S20-S22 a5-d6). We can compare the Zhangjiakou station in
the NCP region with the Hongyuan station in the SB region, and find that the wind speeds at these
stations are almost not overestimated (Fig. 15 b1, a3). The effect of the model on terrain smoothing
contributes to it. Because the wind speed itself increases with the increase of height, and it decreases
when the model smooths over the terrain.
Unlike the 10-m wind speed, the simulation results of the 10-m wind speed have the smallest bias
for the YSU scheme, which is closer to the observed value (Table 2, Figs. 8, 10-11). Of course, this
phenomenon can also be found from the evolution of the wind speed in the vertical direction (Fig.
15, S20-S22). However, as the height increases, the bias of the YSU scheme gradually increases and
is greater than the other three schemes (Fig. 15, S20-S22). Such a large vertical gradient of wind
speed in the YSU scheme indicates a weak mixing in this scheme. From the turbulent diffusion
coefficients of the momentum at 08:00 in January, it is true that the YSU scheme simulates the
smallest turbulent diffusion coefficient below 1000 m (Fig. S23). While the BL scheme simulates a
smallest vertical gradient of wind speed, which corresponds to the largest turbulent diffusion
coefficient of momentum (Fig. S23). The time variation characteristics of the turbulent diffusion
coefficient will be deliberated and analyzed in detail later.
The simulation results of wind direction notes that the model can capture the characteristics of wind
direction well, and it also simulates well for the stations with more complicated topography and



higher altitude (Figs. 15, S20-S22).

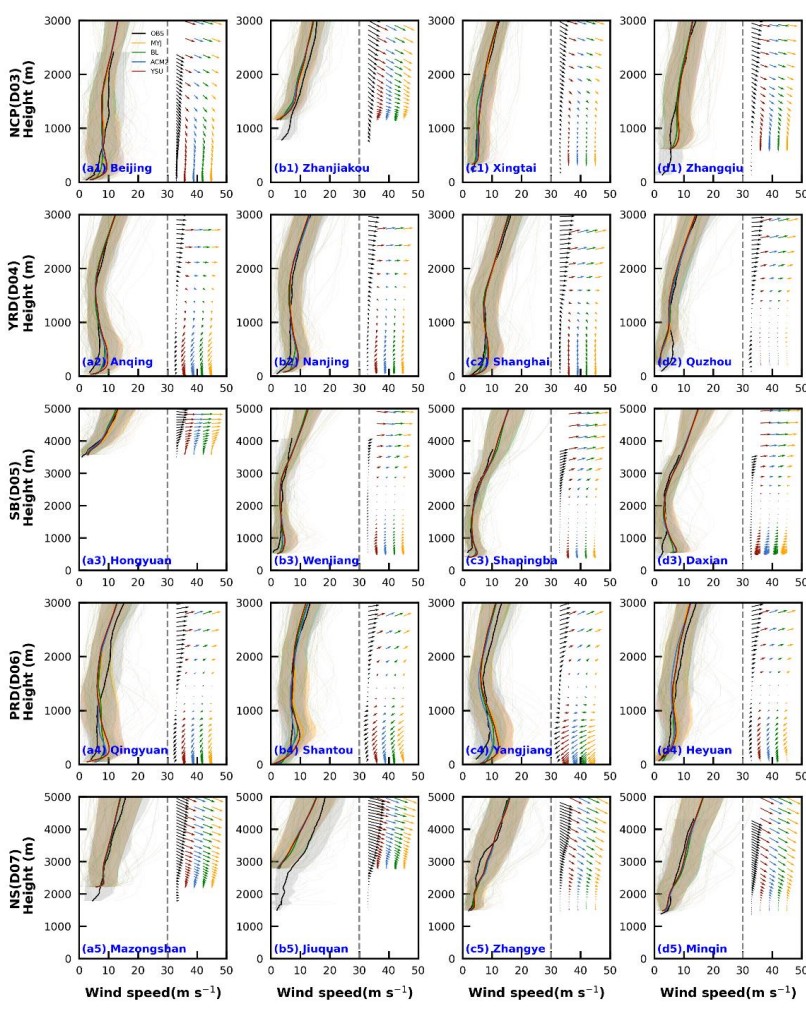


**Figure 15. Similar as Figure 14, but for 10-m wind speed and direction.**

**3.3 PBLH**
Based on the observed data, comparing the PBLH calculated by the two methods, it is found that
the results are mixed for two methods (Fig. 16). The results for January (IOA=0.70~0.90) are better
than the other three months (IOA=0.44~0.88 in April, IOA=0.51~0.86 in July, IOA=0.60~0.77 in
October), and the results in the NCP region are better than the other four regions (Fig. 16). The
PBLH in the NS region are more scattered, unlike the other regions where most of the PBLH are
concentrated below 500 m, especially in April and July (Fig. 16 b5-c5). On the whole, the difference



**811** in PBLH calculated by the two methods is more obvious in the NS region with more complex

**812** topography, which is especially noted when calculating in this type of underlying surface region.

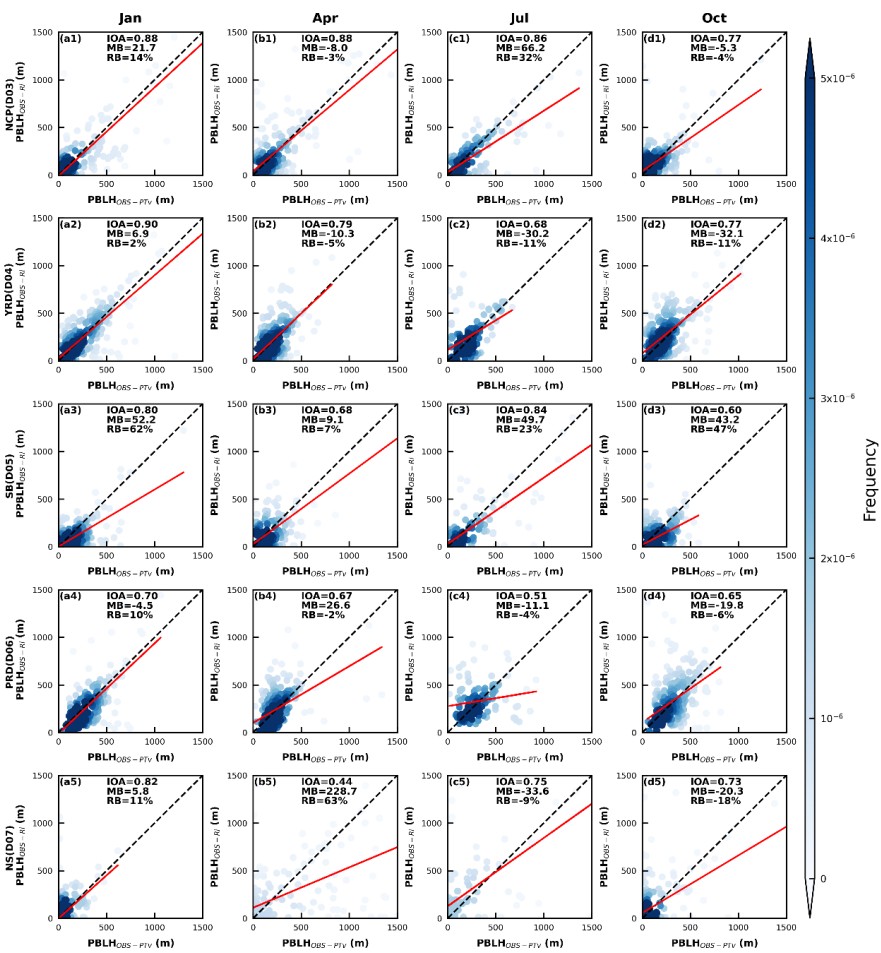

**813**

**814** **Figure 16. Density scatterplots of the PBLH by the two methods in five regions for four seasons.**

**815** Further, the mechanism of understanding the PBLH differences based on different stations from

**816** different regions. The planetary boundary layer height (PBLH) for the YSU and ACM2 schemes is

**817** calculated by the Richardson number method, and the PBLH for the BL scheme is obtained by the

**818** virtual potential temperature method (see Section 2 for details). The same two methods are also used

**819** to calculate the PBLH with sounding data for comparison (Eqs. 29 and 30). First, we compare the

**820** PBLH calculated based on the observed data using Richardson number (Ri) with the PBLH

**821** simulated by YSU and ACM2 schemes. The Ri is determined by both buoyancy term and the shear

**822** term together (Eq. 29). The difference between simulated and observed temperature gradient is

**823** smaller than the wind speed gradient within the PBL (Figs. 14-15). Therefore, the difference in Ri



mainly comes from the variation in the shear term. The wind speed gradient simulated in both
schemes are greater than the observed values (Fig. 15), except for individual stations, which would
result in small values of Ri. Thus, the height of Ri up to 0.25 would be high and the PBLH would
be high. Consequently, the PBLH simulated by the YSU and ACM2 schemes are higher than the
observed values at most stations. For example, in the case of the Quzhou station in the YRD region,
the simulated wind speed gradient at this station is much smaller than the observed value in January,
thus, the simulated PBLH is correspondingly smaller than PBLH calculated from observations (Figs.
15, S24 d2). Comparing the results of the other three months, we can also find similar conclusions
(Figures not shown). The wind speed gradient simulated by the YSU scheme is larger than that of
the ACM2 scheme, and therefore the PBLH is larger than that of the ACM2 scheme, except for the
Shanghai station in the YRD region, Shantou and Yangjiang stations in the PRD region (Fig. S24).
For the ocean, the PBLH simulated by the ACM2 scheme is higher than that of the YSU scheme,
while for most areas adjacent to the ocean, the PBLH simulated by the ACM2 scheme is on the high
side in the YRD and PRD regions (Fig. S25). The simulated PBLH of the BL scheme is in better
agreement with the PBLH calculated by the virtual potential temperature method, which is
substantially better than the other three schemes. The PBLH simulated by the MYJ scheme is mixed
(Fig. 17).



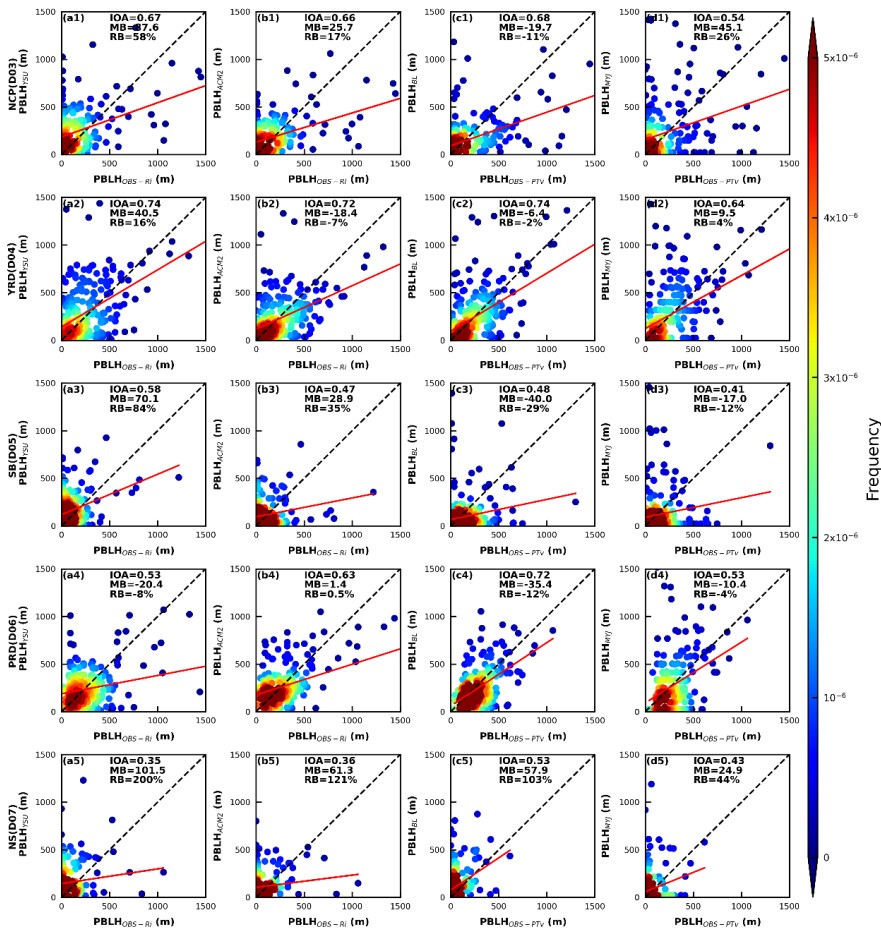

**Figure 17. Density scatterplots of observed and simulated PBLH by four PBL schemes in five regions in January (Winter).**

From the differences of regional distributions, the region with the best PBLH simulation results is the YRD region in January, (IOA=0.64~0.74; MB=-6.4~40.5 m; RB=-2%~16%), followed by the PRD region (IOA=0.53~0.72; MB=-35.4~1.4 m; RB=-12%~0.5%), and the worst simulation results for the PBLH in the NS region (IOA=0.35~0.53; MB=24.9~101.5 m; RB=44%~200%) (Fig. 17). Also, the PBLH simulation in the SB region is poorer and slightly better than that in the NS region, noting that there is still much potential for the model to improve the reproduction of the PBLH in complex terrain. From the simulation results of the four schemes, the PBLH simulated by the BL scheme is the closest to the observed value, followed by the YSU and ACM2 schemes, and the PBLH simulated by the MYJ scheme is the worst in that the simulation results of temperature are better compared with other meteorological factors, so the method of judging the PBLH using the virtual potential temperature will be more consistent with the observed values. While as the YSU



and ACM2 schemes using Richardson number method will involve the wind speed gradient, and
the vertical gradient of wind speed is poorly simulated below 1000 m. That's why it will affect the
judgment of the PBLH. If the simulation results of vertical gradient of wind speed can be improved
subsequently, then the simulation results of PBLH of these two schemes will be improved to some
extent. There are not enough observations to calculate the PBLH using TKE, so there will be some
differences with the PBLH simulated by the MYJ scheme. The mean bias of the simulation increased
in April and July when the PBLH is higher compared to January, with mean bias of -29.6~361.8 m
(6.5~603.9 m), -12.6~410.6 m (41.6~603.2 m), -34.1~301.1 m (3.2~683.9 m) and -14.5~96.3 m (-
11.3~523.6 m) for the YSU, ACM2, BL and MYJ schemes in April (July), respectively. Similar to
January, the best simulation results have been obtained for the YRD (MB=7.8~72.4 m in April,
MB=28.5~66.5 m in July) and PRD (MB=-34.1~-12.6 m in April, MB=-11.3~54.8 m in July)
regions, and the worst for the NS region (MB=61.8~410.6 m in April, MB=523.6~683.9 m in July).
The results for October are more similar to those for January, with lower PBLH and better
simulations than those for April and July (Figures not shown).

### 3.4 turbulent diffusion coefficient

Since the model itself does not directly output turbulent diffusion coefficients for all schemes, there
is relatively little direct comparison and analysis of this parameter. As seen in section 3.3.2 above,
the turbulent diffusion coefficient (TDC) plays a crucial role in the momentum vertical transport
within the PBL and also has an impact on the diffusion of other parameters, such as heat, water
vapor and pollutants (Ding et al., 2021; R. B. Stull, 1988). The accurate portrayal of the TDC directly
affects the evolution of the PBL structure. Based on the contents of section 2.3, the momentum TDC
is not equal to the heat TDC under unstable and neutral conditions for the YSU scheme, and the
momentum TDC is equal to the heat TDC under stable conditions. The ACM2 scheme uses the
MOST method to calculate the TDC as the YSU scheme, but also considers the TDC calculated by
the mixing length theory. The momentum TDC is not equal to the heat TDC in the MYJ scheme,
while the momentum TDC is equal to the heat TDC in the BL scheme. Because of the difference in
altitude of different stations, Bejing station in the NCP region, Nanjing station in the YRD region,
Daxian station in the SB region, Qingyuan station in the PRD region and Zhangye station in the NS
region were selected as representative stations to analyze the turbulent diffusion characteristics.
Here, the TDC of heat is taken as an example, the following basic characteristics have been found.
(1) the YSU and MYJ schemes have the largest TDC during the day, followed by the BL scheme,
and the ACM2 scheme has the smallest TDC (Fig. 18). (2) The TDC is largest in April and July, and
smallest in January and October (Fig. 18). (3) There are significant seasonal differences in the PBLH
for the NCP, SB and NS regions, while for the YRD and PRD regions. The difference in the PBLH
affects the variation of the turbulent diffusion, especially for the YSU and ACM2 schemes, where





the PBLH is used during the calculation of the turbulent diffusion. In the YSU scheme, the TDC of
momentum is calculated first, and then the TDC of heat is calculated with the Prandtl number (Pr).
Thus, the variation of the PBLH is proportional to the TDC (Fig. 18a). While in the ACM2 scheme,
the TDC of heat is calculated directly based on the dimensionless function of heat. Moreover, the
Pr in the YSU scheme varies with height, while the Pr in the ACM2 scheme is a constant (=0.8). It
is also worth noting that in the ACM2 scheme, another TDC is calculated using the mixing length
theory, and the change of the empirical stability function in the mixing length method changes the
TDC. Therefore, the YSU scheme calculates a large TDC of momentum, which also leads to a large
TDC of heat. The TDC of heat in the ACM2 scheme, on the other hand, will be affected by the
mixing length method, and differs from the calculation principle of the YSU scheme.

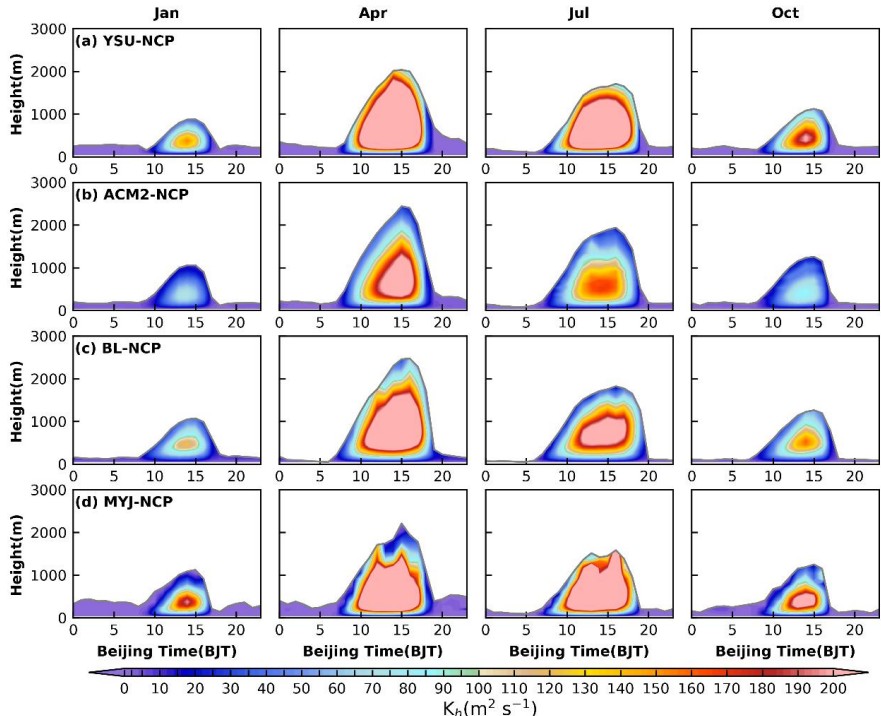


**Figure 18. Time-height cross sections of heat turbulent diffusion coefficient simulated by (a) YSU**
**scheme, (b) ACM2 scheme, (c) BL scheme and (d) MYJ scheme for four seasons in the NCP region.**
**The gray line indicates the PBLH.**
From the section 3.2.2 above, it is clear that the BL scheme has the strongest turbulent diffusion at
08:00, making the vertical gradient smaller, especially the wind speed is large. Similarly, this
phenomenon can be found in the daily variation of turbulent diffusion, and not only at 08:00, but
almost throughout the night (Fig. 18). The difference between MYJ and BL schemes is mainly
reflected in the calculation principle of mixing length, which is not directly related to the PBLH. In





the BL scheme, mixing length scale can be relative to the distance that a parcel originating from this
layer, can travel upward and downward before being stopped by buoyancy effects (Eq. 14).
Therefore, the vertical height below the temperature inversion layer at night, with the surface as the
lower boundary, is the length scale of turbulence, i.e., mixing length scale. While in the MYJ scheme,
the mixing length scale is equal to $z$ minus the integral depth scale, which is equal to the height of
the equal-area rectangle under the profile. It is worth noting that the mixing length scale in the BL
scheme mainly considers the effect of thermal and takes temperature gradient as the criterion, while
the turbulent length scale in the MYJ scheme is mainly determined based on the TKE. TKE is further
divided into horizontal TKE and vertical TKE. Horizontal TKE is mainly influenced by wind shear
and the turbulent eddy scale can reach 1.5~3 times the PBLH on the horizontal, and even reach 6
times the PBLH(Atkinson and Zhang, 1996). The vertical depth of an unstable layer capped by an
inversion is automatically selected as the length scale for turbulence in the BL scheme during the
daytime. Moreover, in the BL scheme, there is a counter-gradient correction term in the convective
PBL, which leads to a downward transport of dry and cool air, making the thermal reach a lower
height and a smaller length scale for turbulence. We also find that the PBLH of the MYJ scheme
exhibits a "sawtooth", and is more pronounced at night. This is mainly because the turbulence is
weaker at night, and presents intermittent characteristics, which, together with the judgment method
of PBLH and the coarse vertical resolution, can cause such variation of the PBLH. Although the
improvement of PBLH in the MYJ scheme cannot have a substantial effect on turbulent diffusion,
the threshold value of its determination method is open to question.
**3.5 Discussion of optimal PBL schemes**
To better understand the simulation performance of different PBL parameterization schemes for
different parameters in each region, this section will discuss the expressiveness of different PBL
schemes through the statistical approach. Figure 19 shows the Taylor statistics for the analysis of
near-surface meteorological parameters in four months in five regions.



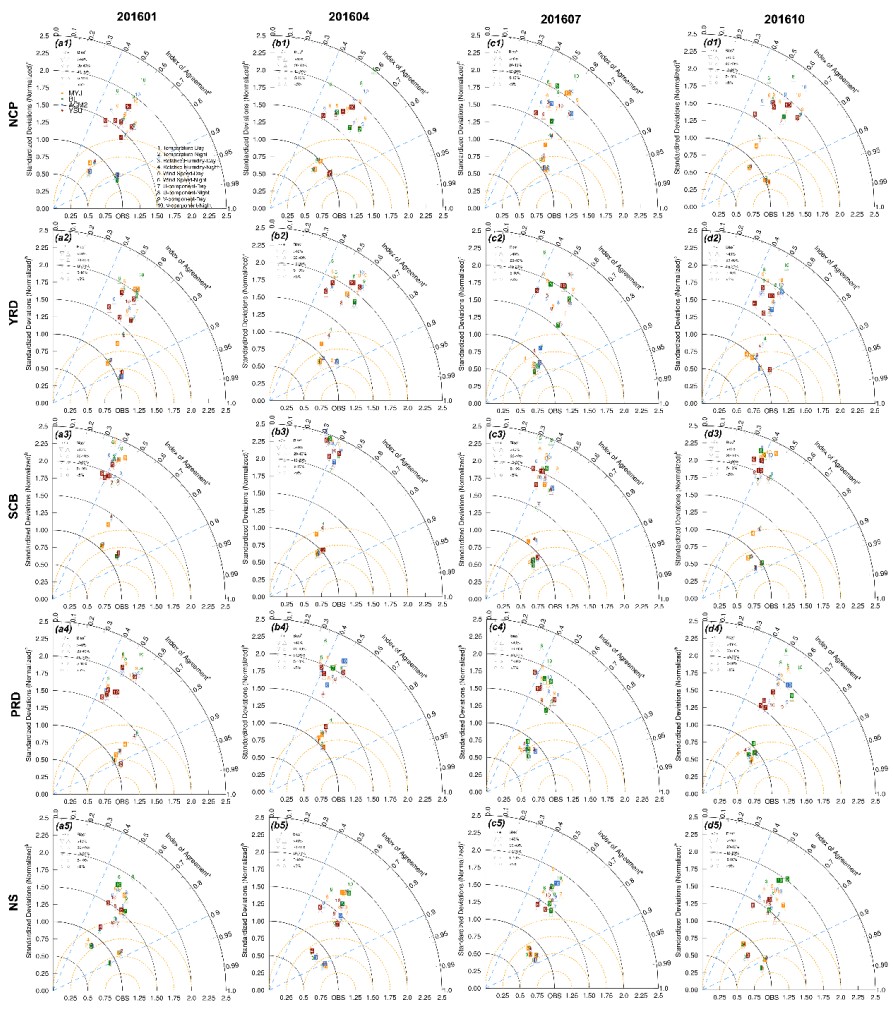


Figure 19. Taylor diagram of observation and simulation in five regions for four seasons. XY axes

and arc represent the normalized standardized deviations (NSDs; NSD =

$\frac{\sqrt{\frac{1}{N-1}\sum_{i=1}^{n}(X_{sim,i}-\overline{X_{sim}})^2}}{\sqrt{\frac{1}{N-1}\sum_{i=1}^{n}(X_{obs,i}-\overline{X_{obs}})^2}}$, $\overline{X_{sim}}$ and $\overline{X_{obs}}$ represent the average value of simulation and observation,

respectively) and index of agreement (IOA; IOA = $1 - \frac{[\sum_{i=1}^{n}|X_{sim,i}-X_{obs,i}|^2]}{[\sum_{i=1}^{n}(|X_{sim,i}-\overline{X_{obs}}|+|X_{obs,i}-\overline{X_{obs}}|)^2]}$, $X_{sim,i}$ and

$X_{obs,i}$ represent the value of simulated and observed, respectively; $i$ refers to time, and $n$ is the

total number of time series), respectively. Four schemes are shown in different colors, and different

numbers represent different parameters. The root mean square is denoted by orange dashed line

and the relative bias (RB; RB = $\frac{\overline{X_{sim}}-\overline{X_{obs}}}{\overline{X_{obs}}} \times 100\%$) is shown by different symbols.



For the NCP region, the 2-m temperatures are underestimated during the daytime in January (Fig.
2), while the BL scheme simulates the highest temperature, so the BL scheme performs optimally.
Although the temperatures are somewhat overestimated at night, the overestimated period is shorter.
The IOA of the four schemes is similar, with the ACM2 scheme having a slightly smaller bias (Fig.
19 a1). Combined with the regional distribution of all stations in the NCP region, the BL scheme is
recommended if the study area is mainly for Beijing, while the ACM2 and YSU schemes are
recommended for the south of the NCP in January, such as Shandong Peninsula and southern Hebei
province. For the other three months, temperatures are overestimated to varying degrees, both
during the day and at night (Fig. 2). The MYJ scheme performs best in all statistical parameters at
night (Fig. 19 b1-d1), while during the daytime, it slightly underperforms the YSU and ACM2
schemes in relative bias in January and April, but the difference is not very distinct. Therefore, for
the simulation of 2-m temperature in other three months, the MYJ scheme would be more
recommended. In the YRD region, the 2-m temperatures are overestimated during the daytime, the
BL schemes show overestimation at night, the MYJ scheme show underestimation, and the YSU
and ACM2 schemes perform optimally (Fig. 2 a2-d2). According to the Taylor statistical parameters,
it can be seen that the ACM2 scheme performs better than the YSU scheme in the four months, and
based on that, the ACM2 scheme is recommended (Fig. 19 a2-d2). The 2-m temperature in the SB
region during the daytime is the same as in the YRD region, and the ACM2 scheme performs
optimally (Fig. 19 a3-d3). However, the BL scheme performs optimally during the nighttime, except
in April (Fig. 19 b3). The PRD region differs from the other regions in that the temperature
simulation is significantly higher in January and April, and the MYJ scheme performs best in both
daytime and nighttime (Fig. 19 a4-b4). In contrast, the temperature simulation bias less in July and
October, and the BL scheme performs best (Fig. 19 c4-d4). The 2-m temperature are almost
underestimated during the daytime and overestimated for the nighttime in the NS region, and the
MYJ scheme outperforms other schemes on account of its large diurnal temperature range (Fig. 19
a5-d5). Of course, the BL scheme presents a slight advantage in the relative bias during the daytime
in January (Fig. 19 a5).
The results of 2-m relative humidity are relatively uniform, and the MYJ scheme shows optimal
simulation performance in almost all months in all regions. Except for July in the YRD region, July
and October in the PRD region, and January and April in the NS region (Fig. 19 c2, c4-d4, a5-b5).
For the simulation of 10-m wind speed and direction, the YSU scheme shows a very clear advantage,
which is outstanding in all regions and all months (Fig. 19).
Several sounding stations with large differences between the observed and simulated altitudes are
removed. Then, the stations in each region are averaged to induce the variation characteristics from
100 m to 2000 m in vertical.



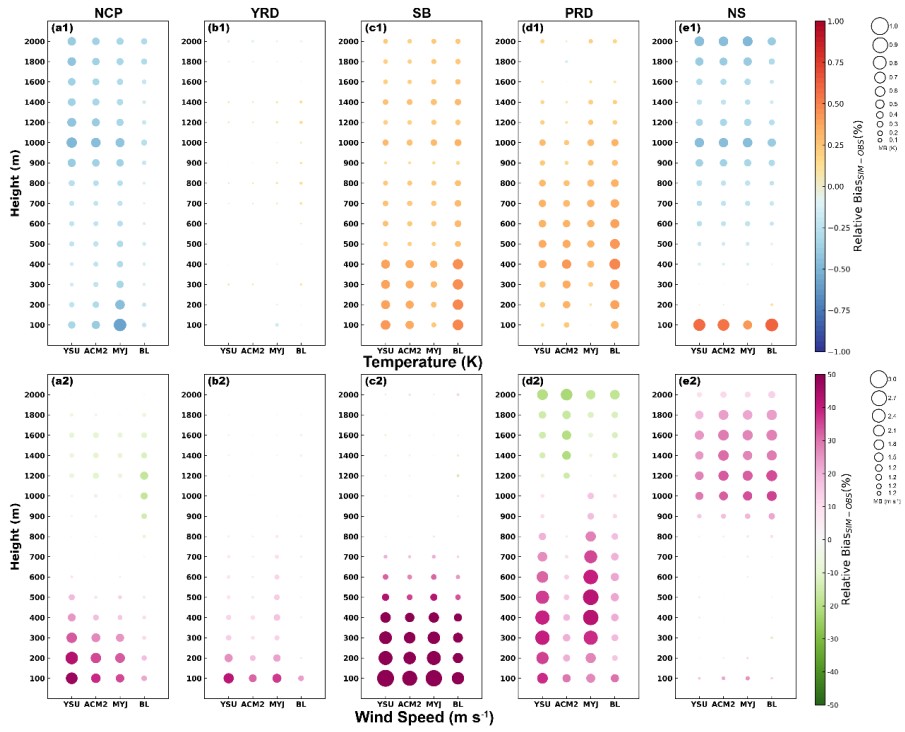

**Figure 20. Statistics of temperature and wind speed in different layers at vertical height in January (Winter), with circle size indicating the mean bias between simulations and observations, and circles filled with color denoting the relative bias.**

The BL scheme has the smallest simulation bias for the temperature in the vertical direction in January for the NCP region, and performs optimally, which is associated with the discussion of the optimal scheme for the 2-m temperature (Fig. 20 a1) to some degree. While in all the other three months, the MYJ scheme has the smallest bias and is consistent with the conclusion of the 2-m temperature (Fig. S26-S28 a1). In the YRD region, there is no clear difference between the four schemes for the simulation of temperature in the vertical direction, and the deviation of the simulation in January is less than 0.1 K (Fig. 20 b1). The optimal scheme for 2-m temperature can be considered as a representative choice. The MYJ scheme has a better simulation of the vertical profile of the temperature that is somewhat different from the most preferred scheme of 2-m temperature in the SB region (Fig. 20, S26-S28 c1). This is mainly because the selected sounding stations are basically located in the basin area with low elevation, and the temperatures are overestimated, as is the 2-m temperature (Fig. 5). If the stations around the basin are not considered, the simulation of 2-m temperature will also be overestimated and the MYJ scheme also perform optimally. There is no complex topography in the PRD region, the results from the sounding stations and surface layer can be well echoed. In the vertical direction, the MYJ simulates the vertical profile

of temperature better, particularly in January and April (Fig. 20, S30, d1). While in July and October,
the simulation results of the four schemes have little difference, especially in the lower level, which
can be represented by the optimal scheme of 2-m temperature (Fig. S27-S28, d1). For the NS region,
there is an overestimation in the lower levels and an underestimation in the upper levels for each
month. The height of overestimation is lower in January and April, around 200 m, while it can reach
around 500 m in July and October (Fig. 20, S26-S28, e1). The positive deviation decreases as the
height increases, but after reaching a certain height, the negative deviation increases again. In the
process of decreasing the positive deviation with height, the MYJ scheme performs the best, which
is consistent with the 2-m temperature. While the BL scheme performs slightly better when the
negative deviation gradually increases with height.
For wind speed, the YSU scheme is optimal for 10-m wind speed for all regions and months.
However, the quite different in terms of the variation of vertical wind speed. Throughout all regions
and months, the simulation bias of the BL scheme is the smallest and closer to the observation,
which is also the results obtained from the section 3.2.2 (Fig. 20, S26-S28 a2-e2). The stronger
turbulent diffusion of the BL scheme at 08:00 makes the wind speed more uniformly mixed in the
vertical direction. And the vertical variation characteristics at 08:00 can be extended to the whole
night. But during the daytime, the result may not be the same, after all, the vertical mixing of the
YSU scheme is stronger, and does not produce such a large wind speed gradient.
In general, in the selection process of the PBL scheme, if the focus is on temperature variation, such
as temperature inversion, the optimal scheme for both 2-m temperature and vertical temperature can
be considered, and there is basically no significant difference. However, if the focus is on the
variation of wind speed, wind energy, then the vertical wind speed and in the surface layer need to
be evaluated and selected with comprehensive consideration.
**4 Conclusion**
The planetary boundary layer serves as a bridge between the ground and the free atmosphere, and
its role cannot be ignored. Turbulence, as the primary motion within the PBL, controls the vertical
mixing of heat, water vapor, momentum, and pollutants. Turbulence as a sub-grid-scale motion is
usually parameterized in the model, i.e., PBL parameterization scheme. The most widely used
mesoscale model (i.e., WRF), which has developed 12 schemes, includes nonlocal closure scheme,
local closure scheme and hybrid nonlocal-local closure scheme. Across the world, there have been
many evaluation studies for PBL parameterization schemes (reference Fig. 1 in Jia and Zhang, 2020).
However, most of the studies have been conducted for individual stations in a small region with
special individual cases for research and analysis, which are not well represented and applied.
Meanwhile, there is a deficiency in understanding the mechanism of the scheme itself. In response,
aiming at the current research deficiencies, four typical schemes (YSU, ACM2, BL and MYJ,





**1032** covering each type scheme) are selected in this study to evaluate and analyze the near-surface

**1033** meteorological parameters, vertical structure of the PBL, PBLH and turbulence diffusion in four

**1034** months (i.e., January, April, July and October) in five typical regions of China (i.e., NCP, YRD, SB,

**1035** PRD and NS regions).

**1036** *a. 2-m temperature.* (1) In terms of time series and diurnal, the simulation results for July are better

**1037** than the other three months, and better at night than daytime, with less deviation between simulation

**1038** and observation. (2) in terms of regional distribution, temperatures at stations with higher elevations

**1039** are easily underestimated (e.g., mountainous areas in the NCP region, areas around the SB basin,

**1040** and the NS region), while overestimated at plains and basin (e.g., YRD, PRD and the SB basin

**1041** regions), and the overestimation/underestimation is more significant during the daytime. (3) In

**1042** terms of mechanism differences between schemes, the differences in the simulated temperatures of

**1043** the four schemes are more pronounced at night. The differences in simulated temperatures between

**1044** the nonlocal scheme mainly originate from downward shortwave radiation, while the effects of

**1045** sensible heat flux (HFX) need to be further ruminated when comparing with the local closure

**1046** scheme. when analyzing the HFX, the gradient of 2-m temperature and surface temperature, the

**1047** variation of heat transfer coefficient need to be discussed in detail.

**1048** *b. 2-m relative humidity.* The changes in relative humidity and temperature correspond to each other,

**1049** and again the best simulation results are obtained in July. Except for the NS region, the relative

**1050** humidity of the other regions is underestimated.

**1051** *c. 10-m wind speed.* (1) The simulation bias is the largest for the MYJ scheme during the daytime

**1052** (except for the NCP region), and the BL scheme presents the largest deviation at night in all regions,

**1053** and the difference is not significant in the four months. The variation of 10-m wind speed is

**1054** influenced by the momentum transfer coefficient, where a larger $C_m$ produces stronger mixing and

**1055** transports more momentum from the upper layers to the lower layers. For the YSU, ACM2 and BL

**1056** schemes, the $C_m$ and 10-m wind speed vary proportionally. In contrast, the MYJ scheme calculate

**1057** principle of for MYJ scheme is different from the other schemes, and the $C_m$ is larger than other

**1058** months almost all day. However, the wind speed simulated by the MYJ scheme is maximum only

**1059** during the daytime, which indicates that it is influenced by integrated similarity functions. (2) In

**1060** terms of regional distribution, the wind speed is more overestimated in plains and basins, and less

**1061** overestimated or even underestimated in mountainous areas. This is chiefly due to the influence of

**1062** the model on terrain smoothing. (3) The overestimation of smaller wind speed at night is more

**1063** obvious in the four schemes, primarily owing to the non-application of the MOST. At night, the

**1064** turbulence intensity is disproportionate to the mean gradient, and the M-O similarity theory is no

**1065** longer applicable.

**1066** *d. 10-m wind direction.* The simulation of wind direction in January for the NCP region worse than

**1067** the other three months, and the frequency of simulated northwest-north winds is overestimated by



about 6.6%. For the YRD region, the frequency of northeasterly winds is overestimated. The
simulation of wind direction in the SB region is not as good as other regions due to the complex
topography. The frequency of northeasterly winds is overestimated in January and October in the
PRD region, and that of southerly winds is overestimated in April and July. The wind direction is
better simulated for the NS region, and the difference is not very obvious.
***e. vertical distribution of PBL.*** The model can reproduce the vertical structure of temperature well,
but the inversion temperature at the lower levels of many stations in complex terrain cannot be
simulated well, mainly because there is a certain difference in the terrain height between observation
and simulation. At 08:00, the MYJ scheme simulates the lowest temperature and the BL scheme for
the highest temperature, and the difference is more conspicuous at the lower levels. The vertical
structure of the wind speed is clearly not as good as the temperature. The wind speed is almost
always overestimated below 1000 m, except for the NS region. Unlike the 10-m wind speed, YSU
has the smallest deviation from the 10-m wind speed, while the BL scheme has the smallest bias in
the vertical direction. The BL scheme has the largest turbulent diffusion and the strongest mixing at

1082    08:00.

***f. PBLH.*** The PBLH calculated based on the observed data using the two methods are better in
January than in the other three months, and in the NCP region than in the other four regions. The
wind speed gradient simulated by the YSU scheme is large, resulting in a small Richardson number
(Ri), making the height higher when Ri reaches 0.25, and the PBLH is higher than that of the ACM2
scheme. The PBLH simulated by the BL scheme is closer to the observation because the temperature
gradient is best simulated. The MYJ scheme results in a jagged variation of the PBLH due to the
determination of the threshold and the vertical resolution, and this phenomenon is especially
obvious at night. In terms of regional distribution, the PBLH is best simulated in the YRD region,
followed by the PRD region and worst in the NS region. The results are similar in January and
October, when the PBLH is lower and the simulations are better than those in April and July.
***g. turbulent diffusion coefficient.*** (1) The TDC simulated by the YSU and MYJ schemes is the
largest during the daytime, followed by the BL scheme, and the smallest by the ACM2 scheme. The
TDC simulated by the BL scheme is the largest at night, and the other three schemes are about the
same. (2) The TDC is maximum in April and July, and minimum in January and October. (3) The
obvious difference in PBLH affects the turbulent diffusion of the YSU and ACM2 schemes. It is
worth noting that the YSU scheme calculates the TDC of momentum first, and then uses Prandtl
number (Pr) to calculate the TDC of heat, while the ACM2 scheme calculates the TDC of both
momentum and heat. (4) The difference between the BL and MYJ schemes is mainly reflected in
the calculation principle of mixing length. The buoyancy effect mainly affects the mixing length
scale in the BL scheme, and the mixing length scale of MYJ scheme is influenced by the TKE.
For the discussion of the optimal scheme, different schemes need to be proposed for different





parameters. (1) Temperature. The BL scheme is recommended for January in the NCP region,
especially for the Beijing, and the MYJ scheme is recommended for the other three months. The
simulation difference between the four schemes is small in the YRD region, and the ACM2 scheme
is recommended. The topography is more complex in the SB region, but the MYJ scheme is
recommended for most areas within the basin, and the BL scheme is recommended for the SB region
if more around basin is involved. The MYJ scheme is recommended for the PRD region in January
and April, and the BL scheme is recommended for July and October. In the NS region, the MYJ
scheme is recommended. (2) Relative humidity. The MYJ scheme is recommended for all regions
in four months. (3) Wind speed. The YSU scheme is recommended if the main concern is the surface
layer, and the BL scheme is recommended if the focus on the variation of wind speed in the vertical
direction.

**Code and data availability**
The source code of WRF version 3.9.1 can be found on the following website:
https://www2.mmm.ucar.edu/wrf/users/download/, and the model settings file is named
"3.9.1_namelist.input", which can be found in the Supplement. In addition, the hourly
meteorological observation data and L-band radiosonde observation data provided by the Chinese
Academy of Meteorological Sciences, are available at https://doi.org/10.5281/zenodo.7792241 (Jia
et al., 2023).
**Author contributions**
Development of the ideas and concepts behind this work was performed by all the authors. Model
execution, data analysis and paper preparation were performed by WJ. XZ and HW provide
computing resources, and offer advice and feedback. YW, DW, and JZ support the data. WZ, LZ,
LG, YL, JW, YY, and YL provides suggestions. All authors contributed to the manuscript.
**Competing interests**
The authors declare that they have no conflict of interest.
**Acknowledgements.**
The work was carried out at the National Supercomputer Center in Tianjin, and the calculations
were performed on TianHe-1 (A).



**Financial support**

This research is supported by NSFC Major Project (42090031), NSFC Project (U19A2044).

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
