# Peer review of "Comprehensive evaluation of typical planetary boundary"

_Geoscientific Model Development, 2023_

## Author Comment (AC1)

**Response to Referee #1**

**RE:** Comprehensive evaluation of typical planetary boundary layer (PBL) parameterization schemes in China. Part I: Understanding expressiveness of schemes for different regions from the mechanism perspective

**Author(s):** Wenxing Jia et al.

In the manuscript "Comprehensive evaluation of typical planetary boundary layer (PBL) parameterization schemes in China. Part I: Understanding expressiveness of schemes for different regions from the mechanism perspective" by Jia et al., the authors select four typical boundary layer parameterization schemes to systematically analyze and evaluate near-surface meteorological parameters, PBL vertical structure, PBLH, and turbulent diffusion in five key regions of China in different seasons. The work can be useful for other researchers to use as a reference when doing simulation studies. I have the following concerns that need to be addressed.

Thank you for your positive comments and valuable suggestions to improve the quality of our manuscript. Based on these comments and suggestions, we have made careful modifications to our pervious draft, and the detailed point-by-point responses are listed below.

**Specific comments:**

(1) Abstract section: The abstract is too long and needs some reduction.

**Re1:** Revised as suggested.

(2) Line 204-207: L-band radiosonde observations by the China Meteorological Administration are generally conducted twice a day (08:00 and 20:00 BJT), except for three times a day at individual stations in summer. More details information about the radiosonde observation data are required.

**Re2:** Revised as suggested.

(3) The convective/unstable boundary layer is an important part of the study of planetary boundary layer. If there is no observation comparison of the daytime PBLH (14:00 BJT), does the author think that the evaluation of PBLH in different seasons in section 3.3 is insufficient.

**Re3:** Thank you very much for your valuable comments! Indeed, we lack the daily trend analysis of PBLH in this section compared to the conventional meteorological parameters, but we have made a basic comparison analysis of the characteristics of different PBL parameterization schemes to capture the PBLH using the data of these two moments. In the future, we will cooperate with other groups to analyze and discuss the daily variation results of PBLH using Lidar data or encrypted sounding data.

(4) Lines 449-451: Please add the equation for calculating sensible heat flux (HFX) here, so that readers can directly understand the relationship between HFX and Ts-T2, and HFX and Ch.

**Re4:** Revised as suggested.

(5) Lines 580-581: Add references to previous studies about wind speed, and explain how these results have been accepted by the public.

**Re5:** Revised as suggested.

(6) Line 591 and 626: The YSU, ACM2 and BL schemes correspond to the revised MM5 near surface scheme and MYJ scheme correspond to another near surface scheme, right? If so, add the near surface scheme corresponding to MYJ scheme in Table 1.

**Re6:** Revised as suggested.

(7) Lines 817-818: Regarding the criterion of the PBLH about the virtual potential temperature method, it seems the PBLH at dawn and dusk of the studied stations is at the transitional stage of the PBL (08:00 and 20:00 BJT). How can the authors prove that the determined PBLH is accurate and can be used to verify the simulation results of the BL scheme?

**Re7:** It may be that we did not express it clearly here, we are comparing the PBLH at 08:00 and 20:00, not the daily average of the PBLH. We have made changes in the text and figures.

(8) Line 819 and 822: Please check that the equation number is correct.

**Re8:** Revised as suggested. Equations 29 and 30 have been modified to equations 15 and 16.

(9) Lines 859-860: For the evaluation of MYJ scheme PBLH, the author uses virtual potential temperature method instead of TKE method. There may be some uncertainties due to different calculation criteria between observation and simulation.

**Re9:** You are right that there is indeed uncertainty in this comparison, and this sentence (L859-860) has been rewritten so that we should point this out in the manuscript.

(10) As the authors mentioned in the manuscript, there is currently insufficient observational data to calculate PBLH using TKE. What observations should be used to calculate PBLH using TKE? What is the calculation?

**Re10:** To calculate the PBLH using the TKE method, the perturbation quantities $u'$, $v'$ and $w'$ of the three components of wind speed are needed, and also the data of different heights in the vertical direction are needed so that the TKE of different heights can be calculated, and then the PBLH can be judged according to the threshold value.

(11) Lines 870-871: This is a new attempt to analyze the turbulent diffusion coefficient. The current simulation results have not yet been verified by observations. While it would be strengthen the paper a lot if observational data can be included and analyzed.

**Re11:** Thank you very much for your acknowledgement that the analysis about turbulent diffusion is currently very scarce, especially from the PBL scheme of the turbulent diffusion mechanism. For now, there are very few observations of turbulent diffusion coefficients, and there is not enough data to be able to analyze them in comparison with the model. We have done some work on the turbulent diffusion of particles with data from one site (Jia et al., 2021a, b; 2022), but the amount of data is

still not enough. In the future, if we can cooperate with other groups to get some observation data and do a special issue on turbulent diffusion coefficient, we hope we can get your attention and correction.

Jia, W., and Zhang, X. Impact of modified turbulent diffusion of $PM_{2.5}$ aerosol in WRF-Chem simulations in eastern China. Atmos. Chem. Phys., 21(22), 16827-16841. doi:10.5194/acp-21-16827-2021, 2021.

Jia, W., Zhang, X.,* Zhang, H.,* and Ren, Y., 2021. Application of turbulent diffusion term of aerosols in mesoscale model, Geophys. Res. Lett., 48, e2021GL093199.

Jia, W., Zhang, X.,* Zhang, H.,* and Ren, Y., 2022. Turbulent transport dissimilarities of particles, momentum and heat. Environ. Res., 211, 113111.

(12) Line 871: "section 3.3.2" should be "section 3.2.2".

**Re12:** Revised as suggested.

(13) Figure 14 and 15: "Zhanjiakou" should be changed to "Zhangjiakou".

**Re13:** Revised as suggested.

(14) Figure 15: The caption is wrong, not 10-m wind speed and direction.

**Re14:** We are very sorry, and we have corrected the title of Figure 15.

(15) Figure 16: What do "Ri", "PTv", "PBLHOBS" and "two methods" stand for? It would be better to state this clearly in the caption.

**Re15:** Based on your comments, we have added detailed descriptions in the corresponding text and figure 16 captions.

(16) Figure 19: The resolution of the image is too low to distinguish the information shown in the image. And "SCB" should be "SB".

**Re16:** In accordance with your comments, we have revised Figure 19 and have checked the entire figure.

---

## Author Comment (AC2)

**Response to Referee #2**

**RE:** Comprehensive evaluation of typical planetary boundary layer (PBL) parameterization schemes in China. Part I: Understanding expressiveness of schemes for different regions from the mechanism perspective

**Author(s):** Wenxing Jia et al.

This manuscript comprehensively evaluates the performance of four typical boundary layer parameterization schemes (YSU, ACM2, BL and MYJ) over four months (January, April, July and October) in five key regions (NCP, YRD, PRD, SB and NS regions) of China, starting from the mechanism of the parameterization schemes to the optimal solution of the parameterization schemes, and answering almost all the answers that current model users want to know in terms of PBL parameterization schemes. The whole manuscript focuses on the near-surface meteorological parameters in the PBL, including 2-m temperature, 2-m relative humidity, 10-m wind speed and direction, vertical structures of the PBL, PBL height, and turbulence diffusion coefficient, analyzes the reasons for the differences between different PBL parameterization schemes, discusses the differences between PBL parameterization schemes and observations, and gives suggestions for the optimal solutions of PBL parameterization schemes. I do have some comments, which I believe should be addressed before the manuscript is considered for publication:

Thank you for your positive comments and valuable suggestions to improve the quality of our manuscript. Based on these comments and suggestions, we have made careful modifications to our pervious draft, and the detailed point-by-point responses are listed below.

**Specific comments:**

1. The abstract section is a bit too long, the article is rich in content and there are many results to show, so you need to take it and streamline the abstract.

**Re1:** Revised as suggested.

2. Why these four boundary layer parameterization schemes were chosen, and whether the corresponding explanatory notes can be given in the appropriate places?

**Re2:** There are three categories of PBL parameterization schemes, nonlocal, local and hybrid. Considering that each category of parameterization schemes should be covered, for each category, a typical parameterization scheme in that category is selected.

3. What was the basis for the selection of sounding sites in each region?

**Re3:** Four sounding stations have been selected for each region, including different underlying surface conditions as much as possible.

4. The second part of the manuscript should describe the topographical and regional characteristics of the five regions and why they were chosen.

**Re4:** Revised as suggested.

5. The analysis of this part of L423-498 is very clear, but the part of L499-522 is a little bit messy and needs to be reorganized, therefore, there is a leading sentence to make the idea clearer.

**Re5:** Revised as suggested.

6. In the section 3.4, are there any observations of turbulent diffusion for the relevant comparison work?

**Re6:** Currently, observations of turbulent diffusion are still scarce, especially for multilayer turbulence observations throughout the vertical direction within the PBL.

7. In fact, there are many articles about PBL parameterization schemes, most of them are more or less the same. This manuscript has analyzed and evaluated the PBL parameterization scheme very comprehensively, which is of great reference to the readers. Could you add a paragraph at the end of the manuscript with thoughts and suggestions on the PBL parameterization scheme?

**Re7:** Revised as suggested. "The PBL parameterization scheme, as the most critical parameterization process within the PBL in the model, has been well proposed and

developed by previous generations, but the development has been slower in recent years, few new theories have been proposed and almost no new schemes have been put into the model or the existing schemes have rarely been improved. Most of the previous studies have evaluated the PBL parameterization scheme, but many of them focus on a particular case in a certain region and lack of universality. This study makes up for this deficiency and provides a comprehensive discussion on the evaluation and uncertainty analysis of the PBL parameterization scheme, hoping to give some reference to the model users. The future development of the PBL parameterization scheme needs to start from the theoretical mechanism, go deeper into the PBL parameterization scheme, and have a deeper understanding of the PBL parameterization, even if it is only for one scheme, or the improvement of one parameter. And for China's self-developed GRAPES model, the introduction and improvement of PBL parameterization schemes need to be selected, rather than a brain to write all the schemes, in fact, many schemes are almost not measured and used."

**Minor issues:**

1. Table 1, A near surface layer parameterization scheme is missing, because MYJ can only couple Eta near surface layer scheme.

   **Re1:** Revised as suggested.

2. L258, "Rib" should be changed to "$Rib_{cr}$".

   **Re2:** Revised as suggested.

3. L331-334, Eq. (13a), Eq. (13b) and Eq. (13c) switch the order and correspond to the previous ones.

   **Re3:** Revised as suggested.

4. L391, "structure" should be changed to "structures".

   **Re4:** Revised as suggested.

5. L400, "orange dots" should be changed to "purple dots" in Figure 1.

   **Re5:** Revised as suggested.

6. L449-451, The formula of HFX should be given out.

   **Re6:** Revised as suggested.

7. L506, here, should also include a part of Shandong Province, "Hebei province" should be changed to "Hebei and Shandong provinces".

   **Re7:** Revised as suggested.

8. L523-524, this sentence needs to be rewritten.

   **Re8:** Revised as suggested.

9. L560-561, according to Figures 2-6 and supplement, this conclusion should be directed especially to winter, i.e., October and January.

   **Re9:** Revised as suggested.

10. L614, "WS10" should be changed to "$WS_{10}$".

    **Re10:** Revised as suggested.

11. L816, "planetary boundary layer height" should be deleted.

    **Re11:** Revised as suggested.

12. L819, here, it is not these two equations, it should be equations (15) and (16).

    **Re12:** Revised as suggested.

13. L822, "Eq. 29" should be changed to "Eq. (15)".

    **Re13:** Revised as suggested.

14. L837-839, what is the reason for the PBLH simulated by the BL scheme is better than the other three schemes, and the poor simulation results of the MYJ scheme?

   **Re14:** This is because the BL scheme calculates the PBLH by the virtual potential temperature method, and the model gives the best simulation results for temperature.

15. L872, "turbulent diffusion coefficient" should be deleted.

   **Re15:** Revised as suggested.

16. L875, "structure" should be changed to "structures".

   **Re16:** Revised as suggested.

17. L888, lack of figure citation, and figure 18 is only for the NCP region.

   **Re17:** Revised as suggested.

18. L904-928, Add some references as supporting notes.

   **Re18:** Revised as suggested.

19. L935-942, the "SCB" in Figure 19 should be changed to "SB".

   **Re19:** Revised as suggested.

20. L1021, "planetary boundary layer" should be changed to "PBL".

   **Re20:** Revised as suggested.

21. L1073, "vertical distribution of PBL" should be changed to "PBL vertical structures".

   **Re21:** Revised as suggested.

22. There are more figures, which need to be provided in high resolution (especially Figure 19), as well as double-checking the figure title of each figure.

**Re22:** Revised as suggested.

23. The citation format of references in the manuscript needs to be standardized, especially in the introduction.

   **Re23:** Revised as suggested.